# Improvement of *On-Site* Sensor for Simultaneous Determination of Phosphate, Silicic Acid, Nitrate plus Nitrite in Seawater

**DOI:** 10.3390/s22093479

**Published:** 2022-05-03

**Authors:** Mahmoud Fatehy Altahan, Mario Esposito, Eric P. Achterberg

**Affiliations:** 1GEOMAR Helmholtz Centre for Ocean Research Kiel, 24148 Kiel, Germany; mesposito@geomar.de; 2Central Laboratory for Environmental Quality Monitoring, National Water Research Center, El-Qanater El-Khairia 13621, Egypt

**Keywords:** Griess reagent, nutrients analysis, Kiel Fjord, posphomolybdenum blue method, silicomolybdenum blue method, vanadium chloride reduction

## Abstract

Accurate, *on-site* determinations of macronutrients (phosphate (PO_4_^3−^), nitrate (NO_3_^−^), and silicic acid (H_4_SiO_4_)) in seawater in real time are essential to obtain information on their distribution, flux, and role in marine biogeochemical cycles. The development of robust sensors for long-term *on-site* analysis of macronutrients in seawater is a great challenge. Here, we present improvements of a commercial automated sensor for nutrients (including PO_4_^3−^, H_4_SiO_4_, and NO_2_^−^ plus NO_3_^−^), suitable for a variety of aquatic environments. The sensor uses the phosphomolybdate blue method for PO_4_^3−^, the silicomolybdate blue method for H_4_SiO_4_ and the Griess reagent method for NO_2_^−^, modified with vanadium chloride as reducing agent for the determination of NO_3_^−^. Here, we report the optimization of analytical conditions, including reaction time for PO_4_^3−^ analysis, complexation time for H_4_SiO_4_ analysis, and analyte to reagent ratio for NO_3_^−^ analysis. The instrument showed wide linear ranges, from 0.2 to 100 μM PO_4_^3−^, between 0.2 and 100 μM H_4_SiO_4_, from 0.5 to 100 μM NO_3_^−^, and between 0.4 and 100 μM NO_2_^−^, with detection limits of 0.18 μM, 0.15 μM, 0.45 μM, and 0.35 μM for PO_4_^3−^, H_4_SiO_4_, NO_3_^−^, and NO_2_^−^, respectively. The analyzer showed good precision with a relative standard deviation of 8.9% for PO_4_^3−^, 4.8% for H_4_SiO_4_, and 7.4% for NO_2_^−^ plus NO_3_^−^ during routine analysis of certified reference materials (KANSO, Japan). The analyzer performed well in the field during a 46-day deployment on a pontoon in the Kiel Fjord (located in the southwestern Baltic Sea), with a water supply from a depth of 1 m. The system successfully collected 443, 440, and 409 *on-site* data points for PO_4_^3−^, Σ(NO_3_^−^ + NO_2_^−^), and H_4_SiO_4_, respectively. Time series data agreed well with data obtained from the analysis of discretely collected samples using standard reference laboratory procedures and showed clear correlations with key hydrographic parameters throughout the deployment period.

## 1. Introduction

Macronutrients such as phosphate (PO_4_^3−^), nitrate (NO_3_^−^), and silicic acid (H_4_SiO_4_) play key roles in the regulation of ocean productivity and thus the marine biogeochemical carbon cycle. In particular, PO_4_^3−^ and NO_3_^−^ are the bioavailable forms utilized by phytoplankton and autotrophic bacteria [1,2], H_4_SiO_4_ exerts a strong influence on the productivity of silicifying phytoplankton such as diatoms, which are estimated to account for 40% of the total primary production in the oceans [3,4]. However, excessive input of PO_4_^3−^ and NO_3_^−^ into estuaries and coastal waters leads to eutrophication, deoxygenation, and other processes that damage aquatic environment [5]. In the open ocean, oligotrophic regions are subject to N and P limitation, which restricts biological productivity [6]. In tropical and subtropical regions, H_4_SiO_4_ is depleted to low levels of ≈0.6 μM, which limits the diatom productivity and thus carbon export from the surface mixed layer [7]. To study these biogeochemical processes, real-time and long-term monitoring of macronutrient concentrations is required to determine the spatial trends and temporal variations in their distributions [8].

Nutrient data obtained from discrete samples usually collected at operational intervals and analyzed using laboratory techniques based on automated colorimetric approaches or ion chromatography. However, such methods are labor intensive, expensive, and yield datasets with a low temporal and spatial resolution [9].

Therefore, there is an urgent need for technologies that enable *on-site* measurements for long-term monitoring and are equipped to cope with challenging conditions during sporadic and transient environmental events. In the last 20 years, a number of studies have been conducted on *on-site* monitoring of nutrients in marine waters [10,11] using mainly three analytical approaches: optical, electrochemical, and wet chemical techniques. In particular, various ultraviolet (UV) optical sensors for routine measurement of NO_3_^−^ have been developed and deployed on different platforms [12,13]. These systems do not require chemical reagents, can measure over a wide range of concentrations, and are easy to use due to their small size and robustness [14]. Optical UV sensors have shown promise for long-term in situ deployment, but their application is limited by low sensitivity and accuracy due to optical interfering factors such as bromide and dissolved organic material [15].

Electrochemical techniques for nutrient measurements facilitate sensor miniaturization, require low power, and in some cases eliminate the need for reagents. Two electrochemical sensors have been reported for H_4_SiO_4_ [16,17,18] and PO_4_^3−^ [19,20]. In these sensors, molybdate (MoO_4_^2−^) ions are introduced into a working solution (NaCl solution (34.5 g L^−1^)) by electrochemical oxidation of a solid Mo wire. Then, either a silicomolybdate or a phosphomolybdate complex is electrochemically produced on an Au working electrode using cyclic voltammetry or square wave voltammetry. Although a short period of a few minutes is required for the electrochemical measurements, a longer period of 30 min is required for PO_4_^3−^ measurements [14]. These techniques seem promising for long-term deployment due to absence of liquid reagents, but further development and investigation is needed for field applications.

Wet chemical methods, also known as reagent-based colorimetric methods, have been used in several *on-site* sensors deployed in rivers, estuaries, coastal waters, and oceans. These methods involve the formation of a light-absorbing dye that provides a robust measurement tool for nutrients with low detection limits and good precision. Among the more recent technologies used for in situ monitoring based on colorimetric assays is the implementation of microfluidics in lab-on-a-chip devices (LOC) [21,22]. Although the LOC technology has shown better performance in terms of lower reagent consumption, lower power consumption, and smaller size compared to other commercially available in-situ analyzers, a multiparameter instrument LOC is not available, and the cost of sensors is relatively high.

Several colorimetric sensors based on flow injection analysis (FIA) have been reported. A submersible chemical analyzer known as Analyseur Chimique In Situ (ALCHIMIST) was installed on a remotely operated vehicle for in situ determination of Σ(NO_3_^−^ + NO_2_^−^) and total dissolved sulfide [23]. NAS2E was used for monitoring of Σ(NO_3_^−^ + NO_2_^−^), and the NH4-Digiscan in situ analyzer was used for monitoring ammonium (NH₄⁺) in coastal and estuarine waters. Other commercially available colorimetric in situ sensors and systems include the Autonomous Profiling Nutrient Analyzer (APNA) and ChemFIN (SubChem Systems, Inc., Narragansett, RI, USA) for NO_3_^−^ and Σ(NO_3_^−^ + NO_2_^−^) analysis, and HydroCycle (Sea-Bird Scientific, Philomath, OR, U S) for PO_4_^3−^. Other systems are based on either the micro loop flow analysis (µLFA) (WIZ, SYSTEA S.p.A., Anagni, Latium, Italy) [24,25] or reverse flow analysis such as the autonomous nutrient analysis in situ (ANAIS) [26].

Recently, new paper-based microfluidic devices for the determination of macronutrients in natural waters have been reported [27,28]. The techniques are based on fluid flow through paper by capillary action without the need for a pump. In principle, the device consists of a sample port into which the water sample is introduced and transport channels connecting other parts of the device, such as the reaction zone, where the analyte solution mixes or reacts with the reagents. The signal (i.e., color formation) is subsequently formed in the detection zone and can be quantified using a cell phone or desktop scanner. Although the proposed systems offer promising applications for *on-site* observations of nutrients in natural waters, the technique does not allow for autonomous continuous monitoring.

All reported wet chemical in situ analyzers are designed to observe single nutrient, and therefore cannot perform multinutrient analysis with the same instrument. An exception is the WIZ probe, but there are no reports in the literature of long-term field testing of these multi-nutrient sensors.

FIA systems based on a single syringe pump and a multiposition switching valve are excellent at compensating for the shortcomings of the continuous flow analyzers currently in use [29,30,31], as they are capable of delivering a small volume (at a level of 10 µL) of reagent without using peristaltic pumps [32]. Automated syringe pump FIA instruments have been developed by EnviroTech LLC (Chesapeake, VA, USA) for *on-site* DNA in situ determination of Σ(NO_3_^−^ + NO_2_^−^), PO_4_^3−^, as well as H_4_SiO_4_ based on the Griess reaction [33] using a Cd column as the reducing agent for NO_3_^−^, the classical blue phosphomolybdate method [34], and the classical silicomolybdate method [35]. The instruments perform routine chemical analyzes according to a preloaded protocol stored in their firmware. However, the protocols show a poor performance and precision, which limits their use for environmental applications in the field. In the stored protocol, only one standard was used for each nutrient. There is no matrix effect correction (i.e., no optical correction) in the sample concentration calculation, as described in the Data Processing Protocol section of the User’s Guide, which limits the use of the analyzer in field deployments. The conventional cadmium column reduction procedure for nitrate determination, which requires regular regeneration, and the rate at which reagents and standards are consumed per measurement, also limit its use for long-term field use.

To the best of our knowledge, there are very few studies that have demonstrated multi-macronutrient analyzers for long periods of deployment. In the present work, we improve the performance of such an instrument by implementing a new nitrate method that uses vanadium chloride (VCl_3_) for the reduction of NO_3_^−^ to NO_2_^−^. This method has been used for a decade in flow analyzers for *on-site* monitoring of nitrate in natural waters [36,37,38]. It showed more promising performance for long-term use than the classic copper-coated cadmium column or zinc particles reported by Ellis et al. in 2011, which must be replaced daily due to reduction efficiency degradation [39]. It also reduces the reagent consumption, which increases the endurance of the sensor for longer deployment. The optimized method was tested during a deployment in coastal waters of the Kiel Fjord, Germany. The new method is validated by additional discrete sampling during the deployment and analysis using a reference air segmented flow analyzer.

## 2. Materials and Methods

### 2.1. Reagents and Standards Preparation

The reagents used in this study were analytical-grade salts prepared with deionized water (resistivity >18.2 MΩ-cm, Milli-Q, MilliporeSigma, Burlington, MA, USA). All glass and plasticware were routinely cleaned, rinsed with deionized water, soaked in 1 M HCl (37%, Carl Roth, Karlsruhe, Germany) for more than 24 h, rinsed with deionized water, and stored in plastic bags before use.

The reagents for PO_4_^3−^ determination were prepared as follows.

-The acidic MoO_4_^2−^ reagent (R1) was prepared by dissolving 12.8 g ammonium molybdate tetrahydrate ((NH_4_)_6_Mo_7_O_24_ 4H_2_O, Sigma Aldrich, Burlington, MA, USA) and 140 mL sulfuric acid (H_2_SO_4_, 98%, Merck, Kenilworth, NJ, USA) to obtain a concentration of 2.57 M (pH 0.6), 3.5 mL of a solution of potassium antimony (III) oxide tartrate trihydrate (PAT; C_8_H_4_K_2_O_12_Sb_2_ 3H_2_O; Merck) (5.3 g/100 mL deionized water), and 1 mL of solution of sodium dodecyl sulfate (C₁₂H₂₅OSO₂ONa; Merck, Kenilworth, NJ, USA) (30 g/L) in 1000 mL deionized water.-Ascorbic acid reagent (R2) was prepared by dissolving 25 g of L(+)-ascorbic acid (C_6_H_8_O_6_; ≥99%, Carl Roth, Karlsruhe, Germany) in 1000 mL of deionized water.

The reagents for H_4_SiO_4_ determination were prepared as follows:-The MoO_4_^2−^ reagent (R1) was prepared by dissolving 15 g of ammonium molybdate tetrahydrate, 5.4 mL of H_2_SO_4_, and 1 mL of sodium dodecyl sulfate solution in 1000 mL of deionized water.-The oxalic acid reagent (R2) was prepared by dissolving 50 g of oxalic acid dihydrate (C_2_H_2_O_4_.2H_2_O; ≥99%, Carl Roth, Karlsruhe, Germany) into 1000 mL of deionized water.-The ascorbic acid reagent (R3) was the same as that used for PO_4_^3−^ determination.

The reagents for NO_3_^−^ and NO_2_^−^ determination were prepared as follows:-The Griess reagent and VCl_3_ reducing agent reagent were prepared by dissolving 5 g of VCl_3_ (Sigma Aldrich, Burlington, MA, USA) in 200 mL of deionized water until the solution turned a dark brown color. Then, 15 mL of concentrated HCl (37%, trace-metal grade, Fisher Scientific, Waltham, MA, USA) was added. After a dark-turquoise color appeared, 10 g of sulfanilamide (H_2_NC_6_H_4_SO_2_NH_2_; Merck, USA) was added by dissolving 1 g of *N*-(1-naphthyl) ethylenediamine dihydrochloride (C_10_H_7_NHCH_2_CH_2_NH_2_.2HCl; Merck, Kenilworth, NJ, USA) and 1 mL of a solution of Triton x-100 50% (*v*/*v*) (50 mL Triton x-100 (Sigma Aldrich, Burlington, MA, USA): 50 mL isopropanol (Fisher Scientific, Waltham, MA, USA) in 1000 mL of deionized water.

Stock solutions of PO_4_^3−^ (1 mM) were prepared by dissolving 0.136 g of potassium dihydrogen sulfate (KH_2_PO_4_; Merck, Kenilworth, NJ, USA) into 1000 mL of deionized water. Stock solutions of H_4_SiO_4_ (1 mM) were prepared by dissolving 0.0212 g of sodium metasilicate pentahydrate (Na_2_SiO_3_.5H_2_O; Sigma Aldrich, city, state, USA) into 1000 mL of deionized water. Stock solutions of NO_3_^−^ (1 mM) were prepared by dissolving 0.0849 g of sodium nitrate (NaNO_3_; Merck, Kenilworth, NJ, USA) into 1000 mL of deionized water. Stock solutions of NO_2_^−^ (1 mM) were prepared by dissolving 0.0689 g of sodium nitrite (NaNO_2_; Merck, Kenilworth, NJ, USA) into 1000 mL of deionized water.

Standard PO_4_^3−^, H_4_SiO_4_, NO_3_^−^, and NO_3_^−^ calibration solutions were prepared by further diluting the respective stock solutions with deionized water.

All reagent solutions were stored in brown 500 mL high-density polyethylene (HDPE) laboratory-grade bottles (Nalgene, Thermo Scientific, Waltham, MA, USA) and kept refrigerated when not in use. Blank, standard, and cleaning solutions were freshly prepared prior to field use and stored in 1000 mL HDPE Nalgene bottles.

### 2.2. Multinutrient Analyzer Description

The analyzer (AutoLAB, EnviroTech LLC, Chesapeake, VA, USA) was a multichannel *on-site* portable chemical analyzer that automatically measures the concentrations of nutrients (Σ(NO_3_^−^ + NO_2_^−^), PO_4_^3−^, and H_4_SiO_4_) in natural waters using wet chemical techniques with colorimetric detection. The system consisted of four main parts, namely, a 16-way rotary valve, a stepper motor-driven syringe, 3 colorimetric detectors, and an electronic controller in a single housing (Figure 1). The rotary valve and the syringe (≈2.2 mL full motion) were driven by a stepper motor controlled by an internal program stored on a memory card and displayed on a terminal interface (Tera Term). A blank, sample, or standard is collected by the analyzer by retracting the syringe plunger while the rotary valve is at the inlet position. Switching the rotary valve and retracting the plunger allows the reagent to be added to the analyte, causing a chemical reaction. This changes the color of the solution contained in the syringe according to the concentration of the nutrient.

The colorimetric detector consisted of a narrow capillary flow cell made of high-grade glass (1 cm path length for the Σ(NO_3_^−^ + NO_2_^−^) and PO_4_^3−^ detector and 2 cm path length for the H_4_SiO_4_ detector with a light-emitting diode (LED) as the light source on one side and a photodiode detector on the opposite side. An additional monitoring photodiode was positioned next to the LED to monitor the intensity of the light source. Green LED with a peak wavelength of 567 nm and a silicon photodiode with a peak intensity at a wavelength of 570 nm were used for Σ(NO_3_^−^ + NO_2_^−^). No information on LED or photodiodes of PO_4_^3−^ or H_4_SiO_4_ detectors was given in the operating manual. To minimize light interference from the outside, the colorimeters were encapsulated in polyurethane. Inside the electronics housing was a series of electronic modules: the main control unit and the motor drivers and detector interfaces. Both the motor drivers and detector interfaces had their own microprocessors and were controlled by the main control unit via a link. Four devices (syringe motor, valve motor, phosphate detector, and (nitrate + nitrite) and silicic acid detectors) were configured through an arrangement called a serial peripheral system (SPS), where the detectors and motors are referred to as SPS devices, and each device had its own SPS address, which is called in the internal scripting language.

The syringe had a polyetheretherketone (PEEK) plunger in a glass cylinder that was fitted with an O-ring. The valve was made of PEEK with a linear polytetrafluoroethylene (PTFE). The swivel fittings were provided with barbed adapters to connect the pump tubing (Tygon LMT-55; green-green, inner diameter 1.85 mm) for fluid transfer. The 0.5 mm PTFE tubing and 1/4 28″ fittings were used to connect the valve to the detector. The same tubing and fittings were used for the sample, connecting via a 1/4 28″ Luer adapter (female–male) PEEK.

### 2.3. Chemical Methods

#### 2.3.1. Phosphate Chemical Assay

The conventional blue method was employed here to quantify PO_4_^3−^ involving a direct reaction with orthophosphate in an acidic MoO_4_^2−^ solution in the presence of PAT to form the yellow phosphomolybdate complex H_3_PO_4_ (MoO_3_)_12_. This solution was then reduced by ascorbic acid as a reducing agent to form the deep blue-colored phosphomolybdate complex [H_4_PMo_8_^(VI)^Mo_4_^(V)^O_40_]^3−^, with extinction measured at a wavelength of 880 nm.

H_4_SiO_4_ has the same tendency to react with MoO_4_^2−^ to form a silicomolybdate complex that adsorbs at 880 nm, interfering with PO_4_^3−^ analysis in seawater. A pH of 0.4–0.9 with a proton/molybdate (H^+^/MoO_4_^2−^) ratio of 60–80 minimizes the interference of H_4_SiO_4_ in the analysis of PO_4_^3−^ [40,41].

#### 2.3.2. Silicic Acid Chemical Assay

The determination of the H_4_SiO_4_ is similar to that of PO_4_^3−^. In particular, it is based on the reaction of H_4_SiO_4_ with MoO_4_^2−^ under acidic conditions of pH 1.5–2 to form the yellow complex H_3_SiO_4_ (MoO_3_)_12_ after a complexation time of 180 s. The solution is then reduced by ascorbic acid in the presence of oxalic acid, which acts as a masking agent for PO_4_^3−^ to form a deep blue colored product with maximum absorbance at a wavelength of 880 nm.

#### 2.3.3. Nitrate and Nitrite Chemical Assay

The determination of NO_3_^−^ and NO_2_^−^ is based on the reduction of NO_3_^−^ to NO_2_^−^ using VCl_3_ at elevated temperatures (≈50 °C) for 30 min. The reduced NO_3_^−^ plus NO_2_^−^ originally present in the sample was quantified by using the Griess reagent method. This method is based on the diazotization of NO_2_^−^ with sulfanilamide to form a diazonium salt, which is then reacted with the coupling agent *N*-(1-naphthyl) ethylenediamine dihydrochloride (NED) to form a pink azo dye with maximum absorbance at a wavelength of 540 nm. The mixed reagent (Griess reagent + VCl_3_) allows for the determination of both NO_2_^−^ and Σ(NO_3_^−^ + NO_2_^−^) and thus the calculation of NO_3_^−^ with the same detector. The reaction mixture can be sent to the detector for NO_2_^−^ determination before the heating step, while Σ(NO_3_^−^ + NO_2_^−^) is determined after the reduction and heating steps.

### 2.4. Analytical Protocol

The complete measurement cycle for each nutrient [PO_4_^3−^, H_4_SiO_4_, or Σ(NO_3_^−^ + NO_2_^−^)] begins with a calibration that includes a blank and three mixed standards with known concentrations of PO_4_^3−^, H_4_SiO_4_, and NO_3_^−^ followed by the analysis of the samples. For each nutrient cycle, after analysis of the highest concentrated standard and samples, a solution of 0.1 M NaOH + 0.5 mL L^−1^ 50 Triton X-100 was drawn into the syringe to wash the system and minimize carryover effects. During the washing step, the three detectors were used to assess the cleaning of the analyzer. To prevent carryover between the solutions during sample analysis, the syringe and colorimeter were flushed twice with 2 mL of either the blank or the standard and six times with 2 mL of the seawater sample. For the PO_4_^3−^ measurement, the analytical protocol involved the drawing of the analyte solution and the two reagents into the syringe in a volumetric ratio of 4:1:1. Mixing was performed by four consecutive back and forth movements of the syringe plunger, which allowed for the initial color to develop. Then, the syringe injected the solution into the detector, allowing the color development to fully develop for 180 s. Finally, the light intensity of the color formed was measured. For the H_4_SiO_4_ measurement, the analyte solution was mixed with the three reagents in a ratio of 1:1:1:1. The analyte solution was mixed with the MoO_4_^2−^ reagent in the syringe, and the flow was stopped for 180 s to allow the yellow complex to form before mixing with the other two reagents. Finally, the solution was transferred to the colorimeter for color determination. For NO_3_^−^ and NO_2_^−^ measurement, the analyte solution was mixed with the modified Griess reagent at a volumetric ratio of 2:1. The solution was then passed into the PO_4_^3−^ detector, where it was incubated at an elevated temperature (≈50 °C) for 30 min. The solution was then transferred to the Σ(NO_3_^−^ + NO_2_^−^) detector for colorimetric determination. For a single sample measurement, a total volume of 137 μL of reagents (i.e., mixed molybdate reagent and ascorbic acid) was used for PO_4_^3−^ determination, a total volume of 396 μL of reagents (i.e., molybdate reagent, oxalic acid reagent, and ascorbic acid reagent) for H_4_SiO_4_ determination, and a total volume of 137 μL of Griess reagent containing vanadium chloride for Σ(NO_3_^−^ + NO_2_^−^) determination was required. A schematic diagram of the syringe pump and the rotary valves is shown in Figure 2. The detailed steps for the nutrient measurement protocol are described in Appendix A and Video S1 [42]. 

### 2.5. Data Processing

The absorbance of the blank, standard, and sample was calculated by using the following equation:(1)$$\mathrm{Absorbance}=-log_{10\;}\left(\frac{V}{V^{R}}\times \frac{V_{0}^{R}}{V_{0}}\right),$$ where V is the voltage of the measuring photodiode (intensity of transmitted light) and $V_{0}$ is the voltage of the monitoring photodiode (intensity of incident light) for the analyte solution after color formation, and $V^{R}$ and $V_{0}^{R}$ are the voltages of the measuring photodiode and the monitoring photodiode for the analyte solution before the reagent was added, respectively. A linear regression between the absorbance of the blank and the three standards was assessed after every 10 measurements. The sample concentration (µM PO_4_^3−^, µM SiO_4_^4−^, or µM Σ(NO_3_^−^ + NO_2_^−^)) was calculated by the following equation:(2)$$\mathrm{Concentration}\;\left(µM\right)=\left(A-B\right)/S,$$
where A is the absorbance of the sample, B is the intercept of the linear fit in the absorbance unit (AU), and S is the slope of calibration curve (AU) µM^−1^.

### 2.6. Field Deployment and Discrete Sampling

A field deployment was conducted on a pontoon in Kiel Fjord, southwestern Baltic Sea, Germany, in May–June 2021. The analyzer was housed in a weather-proof aluminum container (Zarges, Weilheim, Germany) that was placed on the pontoon (Figure 3). The analyzer was fed with a continuous water flow from a depth of 1 m using a submerged water pump with an output of 600 L/h and power consumption of 8 W (Eheim, Deizisau, Germany). The pump inlet was protected by a Cu net (mesh size ≈ 0.297 mm). The water flow was diverted to the analyzer’s sample inlet through a 0.45 μm polyethersulfone syringe filter (Millipore). The analyzer was equipped with a blank solution and three standard solutions for NO_3_^−^ (1, 5, and 10 μM), PO_4_^3−^ (0.5, 1, and 2 μM), and H_4_SiO_4_ (1, 10, and 20 μM), all prepared in artificial seawater (17 g L^−1^ NaCl). After every 10 sample measurements, a calibration procedure was performed. A multiparameter sonde EXO2 (YSI, Yellow Springs, OH, USA) was deployed beside the analyzer to monitor salinity, temperature, and dissolved oxygen (DO). The EXO2 Sonde was deployed on 22 May at a depth of 1 m and sampling frequency of 1 min. Discrete samples were collected from the outlet of the pump filtered through a 0.45 µm syringe filter connected to a 60 mL acid-washed plastic syringe into acid pre-washed 15 mL low-density polypropylene tubes (SEAL Analytical Ltd., Southampton, UK) The collected samples were immediately frozen for later analysis using a QuAAtro continuous air segmented flow analyzer (SEAL Analytical Ltd.). Ancillary data such as wind speed, water temperature, rain precipitation, and solar radiation were obtained from the GEOMAR weather station positioned near the deployment site [43].

## 3. Results and Discussion

### 3.1. Optimization of Analytical Conditions

Different analytical conditions were studied to obtain the highest possible sensitivity for the nutrient measurements. The influence of key analytical parameters was evaluated, including reaction time for PO_4_^3−^, complexation time for H_4_SiO_4_, and analyte/reagent ratio for NO_3_^−^. For PO_4_^3−^, the reaction time was identified as the period between the stopping of the flow and the color development of the reaction mixture in the measurement flow cell. The reaction time varied from 0 to 300 s, and the analytical sensitivity was calculated from the absorbance values of the blank solution and two standard solutions (1 and 2 µM PO_4_^3−^) (Figure 4a). With an increase in reaction times, the analytical sensitivity increased from 0.0058 (±0.0025) AU µM^−1^ and an RSD of 43.9% for 0 s to 0.173 (±0.0002) AU µM^−1^ and 1.19% RSD for 120 s. Sensitivity continued to increase at 180 s with a mean value of 0.0181 (±0.0004) AU µM^−1^ and RSD of 2.41%. A slight decrease was observed at 240 s with a value of 0.0173 (±0.0011) AU µM^−1^ and an RSD of 6.22%. The maximum sensitivity was reached at 300 s with a mean value of 0.021 (±0.0045) AU µM^−1^ and an RSD of 21.9%. On the basis of the highest sensitivity value and the lower RSD value (5% level), 180 s was chosen as the optimal reaction time.

For H_4_SiO_4_, the complexation time was identified as the time during which the analyte and Mo reagent reacted in the syringe and thus the time before the addition of the other two reagents (oxalic acid and ascorbic acid). Increased analytical sensitivity was observed with an increase in the complexation time from 0 s (0.0095 ± 0.00042) AU µM^−1^ and an RSD of 4.39% to 300 s (0.0181 ± 0.000436) AU µM^−1^ and an RSD of 2.4%. With an RSD value of 1.54%, and a change in analytical sensitivity (Δ*s*) of 0.0068 AU µM^−1^ from 0 s to 120 s and Δ*s* of 0.0018 AU µM^−1^ from 120 s to 300 s, 120 s was chosen as the optimal time for complexation. This shows good sensitivity with a low RSD for 120 s and no further improvement for complexation times of up to 300 s (Figure 4b).

For NO_3_^−^ measurements, the reaction temperature is crucial [44,45]. We set the temperature to the maximum value (≈50 °C) and tested the reduction time from 20 min to 50 min (Appendix A). An improvement in reduction efficiency was obtained when we increased the reduction time from 20 min (61%) to 30 min (63%), while no further improvement was noted when the reaction time was increased to 50 min. Therefore, 30 min was chosen as the optimal reaction time. The analyte/reagent ratio was also investigated, and the maximum sensitivity was obtained at a ratio of 2:1 with a value of 0.054 AU µM^−1^. The results were plotted against the ratio of absorbances values of NO_3_^−^ and NO_2_^−^ of the same concentration (to obtain reduction efficiency). As shown in Figure 4c, the efficiency gradually decreased from a ratio of 1:1 (69%) to reach 66.8% at a ratio of 1:6, and then increased at a ratio of 2:1 (76.5%) and before decreasing again at a ratio of 4:1 (73.5%). On the basis of these results, a 2:1 ratio of analyte/reagent was chosen as optimal.

### 3.2. Effect of Salinity

Large variations in salinity were observed in estuaries and coastal waters compared to freshwater and open oceans, and these can affect the measurement of nutrients due to matrix differences. The effect of salinity on the analytical sensitivity of colorimetric measurements can be illustrated by two aspects. The first is the difference in refractive index between the saline sample and fresh water due to the salt effect, referred to as the Schlieren effect. The second is the effect of ionic strength on the analytical sensitivity. These effects occur at high salinity when the transmitted light is directed to the monitoring photodiode [46].

When the flow cell is filled with seawater, a lower voltage is measured by the photodiode than with deionized water. As a result, for the same analyte concentration, lower absorbance values were obtained for the samples in a seawater matrix compared to those in the deionized water matrix. Equation (1) was used to calculate the absorbance offset, which was corrected by subtracting this offset from the sample absorbance after color development.

Salinity variations have an effect on the chemistry used for each nutrient species. The Griess reaction, which involves reduction of nitrate based on VCl_3_, is strongly affected by salinity fluctuations [36]. To investigate the influence of the salinity variations on the analytical sensitivity, the determination of a standard solution of 5 µM NO_3_^−^ was used for solutions with different salinity values that were prepared by dissolving different amounts of NaCl in deionized water. Figure 5a shows an absorbance of 5 µM NO_3_^−^ in deionized water S = 0 (0.14 AU), with absorbance values decreasing with increasing salinity from S = 7 (0.105 AU) to S = 14 (0.09 AU). A steady state was reached with increasing salinity to S = 23 (0.09 AU) and to S = 35 (0.09 AU).

The influence of salinity variations on the analytical sensitivity in H_4_SiO_4_ measurements has been reported [47], with a reported molar absorptivity of the silicomolybdate blue complex in distilled water of 22 × 10^3^ L mole^−1^ cm^−1^ and in oceanic seawater of 19 × 10^3^ L mole^−1^ cm^−1^. Figure 5b shows the absorbance of 5 µM H_4_SiO_4_ in deionized water S = 0 (0.07 AU) with a higher value compared to salinities ranging from S = 23 (0.06 AU) to S = 35 (0.005 AU).

The analytical sensitivity of the Mo-Blue method for PO_4_^3−^ is not affected by the variation in the salinity of the sample matrix [48,49]. The Schlieren effect is the bias that generally occurs in the analytical signal when onboard blank and standard solutions with different salinities than the seawater samples used [50]. This is evident when comparing the analytical sensitivity of 1 µM PO_4_^3−^ standard in a solution of deionized water (S = 0) (0.008 AU) with that using a solution of S = 7 (0.008 AU). Although no large bias was observed when comparing samples with different salinities of S = 7–35 (Figure 5c), an RSD value of 2.11% was noted. Variations in salinity had little effect on analytical sensitivity after applying the optical correction based on Equation (1) compared to values obtained without optical correlation (i.e., via the traditional Beer’s law equation $A=-log_{10\;}\left(\frac{V}{V_{0}}\right)$), where V is the voltage of the measuring photodiode (intensity of transmitted light) and $V_{0}$ is the voltage of the monitoring photodiode (intensity of incident light).

We attempted to correct for the salinity error during measurement by taking the photodiode measurement for the analyte solution before addition of the reagents. Appendix A shows the measured concentrations of 5 µM NO_3_^−^ (Appendix A), 5 µM H_4_SiO_4_ (Appendix A), and 1 µM PO_4_^3−^ (Appendix A). The values obtained with the traditional Beer’s law equation are shown as red circles, while those obtained with Equation (1) using the optical correction are shown as black circles. The comparison between the two values showed that the values obtained with the traditional Beer’s law were underestimated by 3.95% (S = 0) for 5 µM NO_3_^−^ compared to values obtained with the optical correction, with the underestimation increasing with increasing salinity to 40.6% (S = 23 and S = 35). An underestimation of 2.5% (S = 0) was found for 5 µM H_4_SiO_4_, increasing to 43.9% (S = 35) with increasing salinity. An underestimation of 1.42% (S = 0) was found for 1 µM PO_4_^3−^, increasing to 16.4% (S = 35) with increasing salinity. Despite the optical correction, it is recommended to use standards with salinity close to that of the studied waters for field work on board.

### 3.3. Analytical Performance

The analytical performance of the analyzer was tested by evaluating a series of calibrations (Figure 6). The calibration plot showed measurements in deionized water spiked with 0.2, 0.5, 0.75, 1, 2, 5, 10, 20, 40, 60, and 100 µM PO_4_^3−^. The calibration plot showed an analytical sensitivity of 0.01211 AU µM^−1^, indicating good linearity over a wide range (0.2–100 µM) of PO_4_^3−^ with a coefficient of determination R^2^ of 0.999. For H_4_SiO_4_, deionized water was spiked with a range of H_4_SiO_4_ standards (0.2, 0.5, 0.75, 1, 2, 5, 10, 20, 40, 60, and 100 µM SiO_4_^−4^). The calibration plot showed a sensitivity of 0.02377 AU µM^−1^ with a good linearity over a wide range (up to 100 μM) of H_4_SiO_4_ with R^2^ = 0.992. For NO_3_^−^ and NO_2_^−^, the calibration plots showed analytical sensitivities of 0.00614 and 0.01202 AU µM^−1^ for NO_3_^−^ and NO_2_^−^, respectively, and broad linear ranges of 0.5–100 µM for NO_3_^−^ and 0.4–100 µM for NO_2_^−^ with R^2^ = 0.998 and 0.995, respectively. The values of the intercepts and slopes of the corresponding calibration curves are presented in Appendix A; standard deviation values, static t-values, and probabilities are also reported, and the data showed that all values were significant (*p* < 0.01), except for the intercept of the calibration curve for silicic acid, because the blank measurements for silicic acid in deionized water showed negative values [51].

The limit of detection (LOD) was calculated as 0.18 μM, 0.15 μM, 0.45 μM, and 0.269 μM, and the limit of quantification (LOQ) was calculated as follows: 0.6 µM 0.3 µM, 1.5 µM, and 0.89 µM for PO_4_^3−^, H_4_SiO_4_, NO_3_^−^, and NO_2_^−^, respectively, where LOD and LOQ were calculated according to IUPAC recommendation [52,53] using the following equations:(3)LOD=3 σ,
(4)LOQ=10 σ,
where σ is defined as the standard deviation of blank measurements (*n* = 10) (blank measurements were made by applying the associated calibration curves of the blank signals).

Technically, the analyzer can detect nitrite in the field, as we described in the Materials and Methods. However, nitrite concentrations in natural waters are typically in the nanomolar range and below our reported LOD, which limits the use of our analyzer to detect nitrite and nitrate separately.

Table 1 indicates the figures of merit of the analyzer following our analytical improvements and compares the performance with other portable *on-site* analyzers reported in the literature and/or commercially available (WIZ [25], APNA [52], Hydrocycle PO4 [53], NAS3X [54], ANAIS [26], ALCHEMIST [23], NuLAB [55], and Lab on Chip (LOC) [56,57,58], as well as other UV spectral sensors for NO_3_^−^ such as SUNA [59], OPUS [13,60], and SUV-6 [61]). The WIZ, APNA, NAS3X, ANAIS, and NuLAB devices are the only multi-nutrient analyzers reported to date. Although they have a number of advantages, there are some limitations to their application in the field. The sensitivity of WIZ analyzers is limited by their high variability at low concentrations. APNA and ANAIS provide four separate units for PO_4_^3−^, H_4_SiO_4_, and NO_3_^−^ or Σ(NO_3_^−^ + NO_2_^−^). APNA and ChemFIN are based on continuous flow injection analysis, in which the sample and reagent are introduced into a carrier stream, resulting in greater dispersion of the sample and affecting long-term sensitivity [62]. ANAIS is based on reverse injection analysis, in which the detection reagent is injected into the mobile phase of the sample, which reduces sample dispersion and ensures high sensitivity over a long period of time. However, this type of FIA still suffers from the fact that detection is performed under non-equilibrium conditions, which reduces sensitivity compared to manual methods [63]. NAS3X and NuLAB are based on the same type of FIA as the AutoLAB method, where a syringe pump and multi-position ports for reagent and sample delivery combine the features of continuous flow analysis with low reagent and sample consumption with the advantages of discrete (batch) sampling and high sensitivity, making them suitable for *on-site* applications [64].

NAS3X with four different units for the measurement of PO_4_^3−^, H_4_SiO_4_, Σ(NO_3_^−^ + NO_2_^−^), and ammonium and NuLAB are limited by the use of a cadmium column for NO_3_^−^ reduction, which limits their application for long-term use as the cadmium column needs to be regenerated regularly to ensure stable analytical efficiency and thus sensitivity. Moreover, cadmium columns are toxic and decompose over time when they come into contact with organic matter in seawater [65].

**Table 1 sensors-22-03479-t001:** Comparison of the AutoLAB (modified) and other available nutrient sensors reported in the literature and that are commercially available.

Analyzer	Method	Linear Range (μM)				LOD (μM)				Ref.
		PO_4_^3−^	NO_3_^−^	NO_2_^−^	H_4_SiO_4_	PO_4_^3−^	NO_3_^−^	NO_2_^−^	H_4_SiO_4_	
WIZ	µLFA ^(a)/^ wet chemistry	0.19–32.2	0.28–71.4	0.15–19.2	-------	0.19	0.28	0.15	------	[25]
APNA, ChemFIN	CFIA ^(b)^/ wet chemistry	0.03–16	0.03–15	0.02–10	0.05–50	0.03	0.03	0.02	0.05	[54]
Hydrocycle, Sea-Bird	FIA ^(c)^/ wet chemistry	0–10	-------	-------	-------	0.075	------	------	------	[66]
NAS3X	FIA/ wet chemistry	0–6	0-300	-------	0–60	0.06	0.05	------	0.06	[67]
ANAIS	rFIA ^(d)^/ wet chemistry	0.1–5	0.1–40	-------	0.5–150	0.1	0.1	------	0.5	[26]
ALCHEMIST	FIA/ wet chemistry	-------	0–40 ^(e)^		-------	------	0.5		------	[23]
Lab-on-Chip	Micro-fluidics/ wet chemistry	-------	0.025–350	0–0.25	-------	------	0.05	0.02	------	[56]
		0.14–10	-------	-------	-------	0.04	------	------	------	[57,58]
		-------	-------	-------	0–400	------	------	------	0.045	[68]
NuLAB	FIA/ wet chemistry	0.2–25	0.2–50 ^(e)^	0.15-35	0.3–60	0.2	0.2	0.15	0.3	[55]
SUNA	UV-spectral	-------	2.4–4000	-------	-------	------	2	-----	-----	[59]
OPUS	UV-spectral	-------	1–60	-------	-------	------	2	-----	-----	[13]
SUV-6	UV-spectral	-------	0–400	-------	-------	------	0.21	-----	-----	[61]
ANESIS	Electro-chemistry	-------	-------	-------	1.63–132.8	------	------	-----	0.32	[69]
AutoLAB (modified)	FIA/wet chemistry	0.2–100	0.5–100	0.4–100	0.2–100	0.18	0.45	0.35	0.15	This work

^(a)^ Micro loop flow analysis, ^(b)^ continuous flow injection analysis, ^(c)^ flow injection analysis, ^(d)^ reverse flow injection analysis, ^(e)^ Σ(NO_3_^−^ + NO_2_^−^).

As part of the evaluation of analytical performance, the accuracy of the analyzer was determined in the laboratory using certified reference material (CRM CG, Kanso Co., Ltd., Osaka, Japan). Ten replicate measurements of CRM CG reference material for nutrients in seawater with an assigned PO_4_^3−^ concentration of 1.7 ± 0.02 µM, assigned NO_3_^−^ of 24.2 ± 0.2 µM and NO_2_^−^ of 0.06 µM, and assigned H_4_SiO_4_ of 57.7 ± 0.5 µM. The means of the measured values were 1.4 ± 0.14 µM, 25.8 ± 2.7 µM, and 49.4 ± 2.8 µM for PO_4_^3−^, Σ(NO_3_^−^ + NO_2_^−^), and H_4_SiO_4_, respectively. The results show that the analyzer is suitable for macronutrient analysis over a wide range of concentrations.

Ten replicate measurements of the CRM were taken over a 10-day period at a frequency of one measurement per day during the deployment to investigate both reproducibility and stability of the analyzer (Appendix A). An RSD of 8.9% was obtained for PO_4_^3−^ with maximum and minimum absorbance values of 0.023 and 0.018, respectively; an RSD of 7.4% was obtained for NO_3_^−^ with maximum and minimum absorbance values of 0.25 and 0.2, respectively; and an RSD of 4.8% was obtained for H_4_SiO_4_ with maximum and minimum absorbance values of 0.58 and 0.5, respectively. The values of RSD are less than the extent reported by Gibbons et al. (10% RSD) [70], showing good precision of the analyzer. These results demonstrate good applicability of the analyzer for the analysis of seawater. The paired *t*-test was used to detect systematic error (bias) at a degree of freedom (df) of 9. No bias was observed for NO_3_^−^ (*t*-value = 2.46, *t*_critical_-value = 2.82, *p* > 0.01), which was not the case for PO_4_^3−^ (*t*-value = 7.95, *t*_critical_-value = 2.82, *p* < 0.01) and H_4_SiO_4_ (*t*-value = 6.163, *t*_critical_-value = 2.82, *p* < 0.01), where there was a significant difference between the assigned values and the measured values. This could have been due to the fact that only one CRM was tested.

### 3.4. Field Deployment

The performance of the analyzer was demonstrated under environmental conditions during a field campaign in Kiel Fjord. The fjord is located on the southwestern coast of the Baltic Sea and is a mesohaline inner coastal water body that is a small extension of the Bay of Kiel. The Kiel Fjord is about 6 km wide at the mouth and has a length of 15 km; its mean and maximum depths are 10 m and 22 m, respectively. The hydrography of the Kiel Fjord is characterized by strong variability in salinity from S = 2.6–22.4 with a mean salinity of S = 14.3 [71]. The higher salinity waters originate mainly from the North Sea, while the lower salinity waters originate from the eastern Baltic Sea with additional riverine inputs. The Baltic Sea is a transition zone between the high salinity water from the Kattegat and brackish water from its own central zone. The salinity in the fjord is strongly influenced by the salinity fluctuations in the Bay of Kiel. The water in the Kiel Fjord is well mixed; during strong wind conditions, the waters can be completely flushed [72]. Temperatures in the fjord range from 0 °C to 22 °C with an annual mean value of 11 °C [71]. Since the tidal range in the Baltic Sea is only 20 cm, the currents at the location of the measurement pontoon along the shore of Kiel Fjord are mainly determined by winds [73,74]. Overall, the water level in Kiel Fjord showed a nearly constant value during the deployment period (12 May to 28 June 2021). Appendix A shows the water level data obtained from the Kiel-Holtenau hydrological station. A mean water level of 502.8 ± 0.04 cm was obtained with minimum and maximum values of 460 cm and 541 cm, respectively. The datasets were obtained from the Federal Waterways and Shipping Administration (WSV) [75].

Figure 7a shows the hydrographic data of salinity, DO, and temperature obtained from the EXO2 Sonde during the period between 28 May and 27 June 2021, with two gaps on 12 June and 11–14 June due to a problem downloading data from the sensor. Water temperature showed a gradual increase from around 10 °C before reaching the maximum of 20.9 °C. Salinity fluctuated during the study period, with minimum and maximum values of 9.7 and 17.08, respectively (mean ± 1 SD; 13.5 ± 1.7). The DO showed a maximum value of 7.4 mg L^−1^ and a minimum of 12.07 mg L^−1^ (mean value of 9.6 ± 0.7 mg L^−1^) throughout the study period (28 May–27 June). Figure 7b shows timeseries data for water temperature obtained using a surface water temperature sensor from 28 May–27 June. Wind speed was obtained from a mast beside the deployment site. Figure 7c shows the time series data for dissolved carbon dioxide concentration (CO_2_ partial pressure (pCO2)) obtained with the CONTROS HydroC-CO_2_ sensor (4H Jena, Germany) mounted at the deployment site at a depth of 1 m adjacent to the sample intake of our analyzer. Two time series were obtained, the first from 3 June to 10 and the other from 17 to 27 June, with a gap in between because the sensor was out of service. For the first period, the mean value was 599 ± 107 µatm with minimum and maximum values of 390 µatm and 1047 µatm, respectively. For the other time period, there was a mean value of 479 ± 70 µatm with minimum and maximum values of 341 µatm and 807 µatm, respectively.

Figure 8 shows the PO_4_^3−^, H_4_SiO_4_, and Σ(NO_3_^−^ + NO_2_^−^) data from the field deployment in Kiel Fjord over 46 days between May 12 and June 27. A total of 443 PO_4_^3−^, 440 Σ (NO_3_^−^ + NO_2_^−^), and 409 H_4_SiO_4_
*on-site* measurements at 66 min intervals was obtained. Outliers were mainly caused by trapped air bubbles in the flow cell and excluded from the time series. Bubbles formed either by clogging of the syringe membrane filter with sediments or by blockage in the copper net with large particles. The effect was evident from the low transmission values measured by the detector for the sample before reagent addition (i.e., before color formation). Appendix A shows the voltage readout of the photodiode detector over the whole period, wherein a reduction in values happened during some periods. One way to avoid fouling of the internal analyzer components problem is to use a tubing with a narrow inner diameter and a slow flow rate (the same approach is used for Sunburst devices [76]). This was not possible with the aquarium pump used in our study as it was not possible to control the flow rate. In future applications, we will use a small pore size (e.g., 1 μm) syringe filter that will prevent internal fouling.

Rainfall data were monitored as the sum of the precipitation over a period of 12 h and measured three times per day at 0, 6, and 18 h UTC (Figure 8, blue lines).

Considering all the *on-site* data, the mean PO_4_^3−^ concentration was 0.26 µM (±0.15); a minimum value of 0.0012 µM (<LOD), and a maximum value of 1.07 µM. The Σ(NO_3_^−^ + NO_2_^−^) concentrations ranged from 0.0025 (<LOD) to 18.6 µM, and the mean was 2.9 µM (±2.3). The H_4_SiO_4_ concentrations ranged from 0.001 µM (<LOD) to 55.9 µM; the mean was 12.2 µM (±10.4). For all data points of discrete samples (Figure 8, red stars), PO_4_^3−^ concentrations were in the range of 0.03–1.11 µM, and the mean was 0.27 ± 0.18 µM (*n* = 51). The Σ(NO_3_^−^ + NO_2_^−^) concentrations were in the range of 0.17–12.6 µM, and the mean value was 1.96 ± 2.51 µM. The H_4_SiO_4_ concentrations were between 0.007 µM (<LOD) and 27.1 µM, and the mean value was 11.1 ± 4.9 µM. As one measurement cycle takes a total of 66 min, comparisons between the *on-site* data and the discrete samples (Appendix A) were made for the data points within a 30 min time interval. For PO_4_^3−^ data points (*n* = 21) (Appendix A), a positive Pearson’s correlation coefficient of 0.6534 was obtained. For Σ(NO_3_^−^ + NO_2_^−^) data points (*n* = 17) (Appendix A), two clear outliers were excluded from the correlation plot. A positive Pearson’s correlation coefficient of 0.45 was obtained. A positive Pearson’s correlation coefficient of 0.47 was determined for the H_4_SiO_4_ data points (*n* = 19) (Appendix A). Although no strong correlation was found between the PO_4_^3−^ data points (in situ vs. discrete samples), there was no significant difference between the means at the 1% level (paired *t*-test, *p*-value = 0.729, df = 20), with the null hypothesis being mean (in situ) = mean (discrete samples). The same was true for the Σ(NO_3_^−^ + NO_2_^−^) data point, where a weak correlation but no significant difference between means was found (paired *t*-test, *p*-value = 0.04, df = 14). For H_4_SiO_4_, the difference between means was also not significant (paired *t*-test, *p*-value = 0.87, df = 18), with both the null and alternative hypotheses similar to those for PO_4_^3−^ and Σ(NO_3_^−^ + NO_2_^−^). The analytical approaches appeared to be reliable to quantify nutrients under environmental conditions, as evidenced by comparison with values reported in the literature for the same time period. Fischer et al. [77] reported the concentration of macronutrients in samples collected at the institute pier at Kiel fjord at a depth of 2 m in May 2011. The mean value of H_4_SiO_4_ was 9.9 µM with maximum and minimum values of 14.5 µM and 5.8 µM, respectively. The mean value of PO_4_^3−^ was 0.3 µM with minimum and maximum values of 0.2 µM and 0.4 µM, respectively. The mean value of NO_3_^−^ was 0.1 µM with minimum and maximum values of 0 µM and 0.2 µM, respectively. Wasmund et al. [78] reported on the concentration of macronutrients in the Bornholm Basin. The Bornholm Basin is located east of the Arkona Basin on the southwestern coast of the Baltic Sea between Sweden and the island of Bornholm. For samples collected in surface waters at a depth of 5 m on 12 May 2016, the average concentrations of NO_3_^−^, PO_4_^3−^, and H_4_SiO_4_ were 0.31 µM, 0.35 µM, and 12.1 µM, respectively.

There are a variety of factors that influence the concentrations and distributions of nutrients in the water column of estuaries (e.g., fjords). The time scales of biogeochemical cycles depend on a variety of conditions, including freshwater inflow from rivers, which in turn depends on the morphology or topographic features of the fjord. Tidal flow controls the input of saline water and mixing processes. The biogeochemical cycles include microbial activity (remineralization), phytoplankton activity, grazing activity by zooplankton, and benthic exchange. In addition, anthropogenic inputs of domestic and industrial waste waters with high nutrient levels strongly affect the concentrations of macronutrients and phytoplankton growth in marine environments of densely populated urban centers [79,80].

The main source of H_4_SiO_4_ in estuarine waters is the weathering of terrigenous rock minerals by naturally acidic rainwater [81]. Phosphorus has important anthropogenic sources (including wastewater), and following biological uptake in the surface waters is removed to subsurface waters and sediments by sinking phytoplankton debris, where it is released following remineralization. Sinking of phosphate associated with iron-oxyhydroxide particles transfers phosphate to sediments, where phosphate is released upon iron (III) reduction to iron (II) under anoxic conditions [82,83], and may be released to the overlying waters. A key source of nitrate to estuarine systems include waste water discharges, but also run-off from agricultural lands of fertilizers [79,84].

The tidal amplitude in Kiel Fjord is low, and hence tidal currents have a low influence on the re-distribution of nutrients, which instead mainly depends on wind-driven processes [73]. Appendix A shows the relationship between the daily average concentrations of macronutrients analyzed *on-site* using the analyzer over the entire deployment period (12 May to 27 June) and the daily average of wind speed. A significant correlation coefficient was obtained for PO_4_^3−^ (r = 0.4, *n* = 37), while two significant correlation coefficients (r = 0.4, *n* = 33) and (r = 0.3, *n* = 31) were obtained for Σ(NO_3_^−^ + NO_2_^−^) and H_4_SiO_4_, with nine points and seven points clear outliers being excluded, respectively. These outliers may be due to the fact that the distribution of nutrients in estuarine water is complicated and may be influenced by various environmental factors rather than just one factor [80].

Remineralization of organic matter in subsurface fjord waters and sediments leads to an increase in pCO_2_ (and macronutrients), with a subsequent transfer to surface waters by wind-driven mixing. As Figure 9 shows, the increase in pCO_2_ during the period from 03 to 10 June with a mean value of 591 µatm resulted in an increased supply of macronutrients through dissolution and respiration processes, leading to a concentration of H_4_SiO_4_ with a mean value of 18.1 µM, a concentration of Σ(NO_3_^−^ + NO_2_^−^) with a mean value of 3.8 µM, and a mean concentration of PO_4_^3−^ of 0.3 µM. For the period from 17 June to 27, a mean value of 472 µatm was obtained for the pCO_2_ value, with a mean concentration for H_4_SiO_4_ of 2.46 µM, a mean concentration for Σ(NO_3_^−^ + NO_2_^−^) of 2.12 µM, and a mean concentration for PO_4_^3−^ of 0.2 µM (except for the clear outlier on 22 June). The slightly increased concentration of PO_4_^3−^ between 17 June and 27 can be explained by the influx of freshwater, with a 2.6 decrease in salinity between the mean salinity from 2 June to 10 (mean salinity of 14.6) and 17 June to 27 (mean salinity of 11.98). Pearson’s correlations were used to evaluate the relationship between pCO_2_ data and the *on-site* macronutrients data, and three significant correlation models (Appendix A) were obtained over 11 days (from 4–9 June, 18 June and 22–27 June) with a correlation coefficient (r) of 0.3 (*n* = 123) between in situ pCO_2_ data and *on-site* PO_4_^3−^ data (*p*-value = 3.2 × 10^−4^), correlation coefficient (r) of 0.3 (*n* = 123) between in situ pCO_2_ data and *on-site* Σ(NO_3_^−^ + NO_2_^−^) data (*p*-value = 8.14 × 10^−4^), and correlation coefficient (r) of 0.3 (*n* = 108) between in situ pCO_2_ data and *on-site* H_4_SiO_4_ (*p*-value = 0.001).

Overall, the time series data demonstrated that the *on-site* nutrient analyzer was able to generate high-resolution data that helped to facilitate our ability to interpret biogeochemical processes of macronutrient cycling, benthic exchange, and water column mixing in Kiel Fjord.

## 4. Conclusions and Future Implications

This work highlights the ability of the AutoLAB multi-nutrient analyzer with optimized analytical protocols to produce real-time, well-resolved measurements of macronutrients in the marine environment. The measurement procedure was improved by changing the measurement sequence, introducing the vandium chloride method for NO_3_^−^ analysis and evaluating the effects of salinity fluctuations. Validations were performed by measurements of CRMs. The deployment in estuarine surface waters of the Kiel Fjord successfully captured the temporal distribution of macronutrients across a period of 46 days; the results were in good agreement with those obtained from the discrete samples analyzed via a laboratory-based air-segmented flow analyzer. Mean concentrations of 0.26 µM for PO_4_^3−^, 2.9 µM for Σ(NO_3_^−^ + NO_2_^−^), and 12.3 µM for H_4_SiO_4_ were measured in the Kiel Fjord from 12 May to 27 June 2021. The analyzer successfully acquired temporal variations via 66 min time sampling intervals. The analyzer was able to provide valuable information that helped to understand the nutrient dynamics of Kiel Fjord waters otherwise poorly captured via the discrete samples collection. The analyzer allowed for the measurement of short-term fluctuations and also monitoring of long-term trends. Environmental variations were confirmed by other sensors placed next to the analyzer at the site.

The LODs for the nutrient analysis by the analyzer are indeed close to those reported in literature or for commercially available systems, but their applicability for long-term *on-site* monitoring of multiple nutrients in natural waters is limited by a range of drawbacks, including:-The option to only determine a single nutrient by an analyzer.-The use of a cadmium column for nitrate reduction, which may degrade by organic matter in the water, and also regular regeneration is typically needed. Our VCl_3_ reduction approach therefore provides an important step forward.-An absence of reports on long-term use or field testing in natural waters for some promising analyzers.

To further test the field application of the multi-macronutrient analyzer, in situ deployments of the EnviroTech LLC’s submersible units (NAS-2E) with the here-developed improved analytical protocol and vandium chloride method for NO_3_^−^ quantification are planned in the near future.

## Figures and Tables

**Figure 1 sensors-22-03479-f001:**
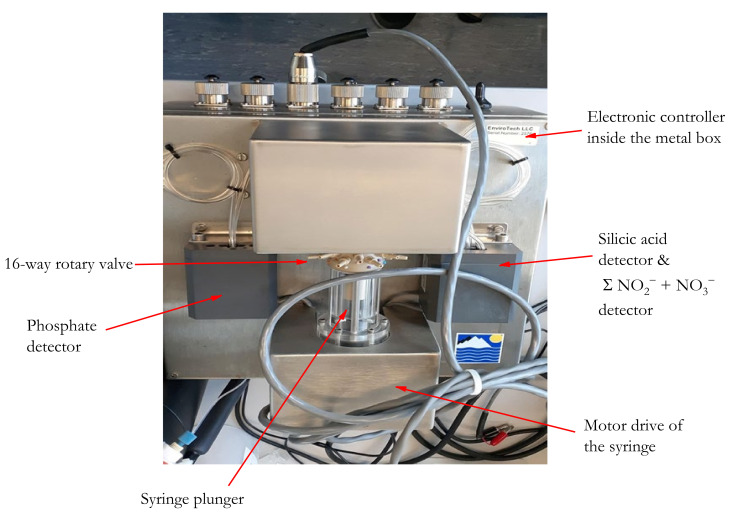
Hardware of multi-nutrient analyzer (AutoLAB) showing the five major components, namely, the 16-way rotary valves, a motor drive of the syringe, the syringe plunger, three colorimetric detectors, and an electronic controller inside a metal box.

**Figure 2 sensors-22-03479-f002:**
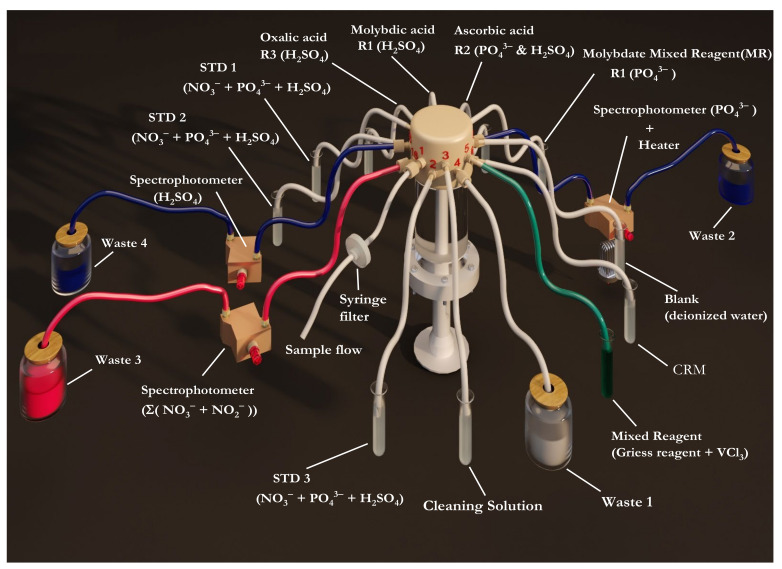
Three-dimensional schematic diagram of the AutoLab autoanalyzer (syringe and 16-way rotatory valve) for multinutrient determination. Standard solutions: STD; certified reference materials: CRM.

**Figure 3 sensors-22-03479-f003:**
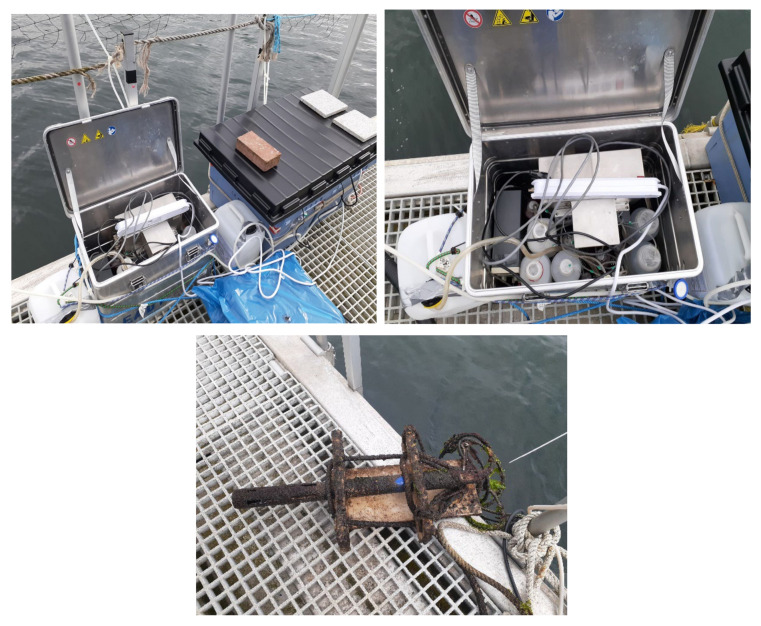
Deployment setup of the AutoLAB analyzer (**upper**) and the EXO2 Sonde after the deployment (**bottom**).

**Figure 4 sensors-22-03479-f004:**
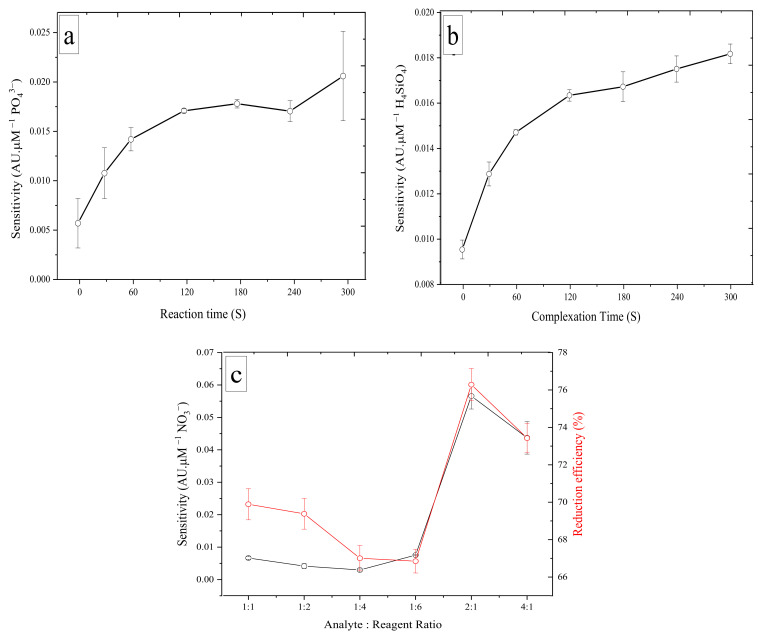
(**a**) Effect of the reaction time on the sensitivity (slope of the calibration curve: 0, 1, 2 µM PO_4_^3−^). (**b**) Effect of the complexation time on the sensitivity (slope of the calibration curve: 0, 1, 2 µM Si). (**c**) Effect of changing the analyte: reagent ratio on the sensitivity of the calibration curve (0, 1, 2 µM NO_3_^−^) (black lines) and on the reduction efficiency (%) (red lines). AU: absorbance unit. Error bar (±1 SD), *n* = 5.

**Figure 5 sensors-22-03479-f005:**
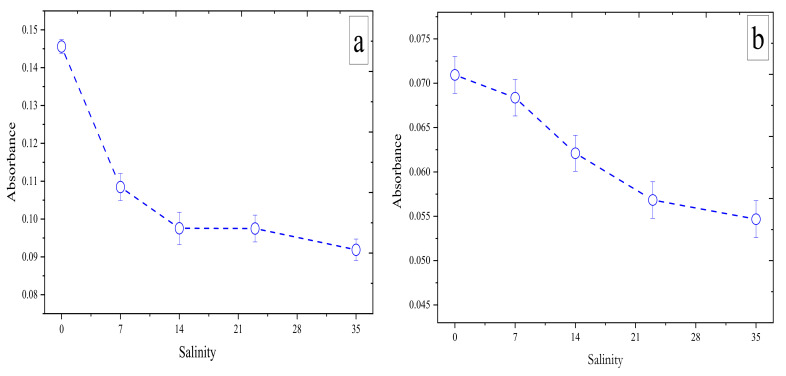
Effects of salinity (S = 0, 7, 14, 23, and 35) on the absorbances of a (**a**) 5 µM NO_3_^−^ standard, (**b**) 5 µM H_4_SiO_4_ standard, and a (**c**) 1 µM PO_4_^3−^ standard. Error bar (±1 SD), *n* = 10.

**Figure 6 sensors-22-03479-f006:**
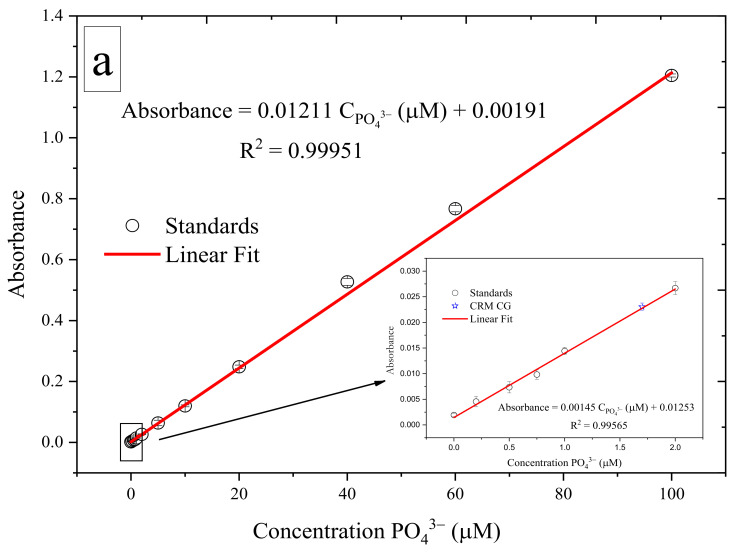
(**a**) Calibration curve for PO_4_^3−^ standards (0, 0.2, 0.5, 0.75, 1, 2, 5, 10, 20, 40, 60, and 100 µM) in a 1 cm flow cell. (**b**) Calibration curve for H_4_SiO_4_ standards (0, 0.2, 0.5, 0.75, 1, 2, 5, 10, 20, 40, 60, 100 µM) into a 2 cm flow cell. (**c**) Calibration for NO_3_^−^ standards (0, 0.5, 1, 2, 5, 10, 20, 40, 60, 100 µM) into a 1 cm flow cell. (**d**) Calibration curve for NO_2_^−^ standards (0, 0.4, 1, 2, 5, 10, 20, 40, 60, 100 µM) into a 1 cm flow cell. Blue stars indicate the absorbance of the Kanso CRM CG. Error bar (±1 SD), *n* = 10.

**Figure 7 sensors-22-03479-f007:**
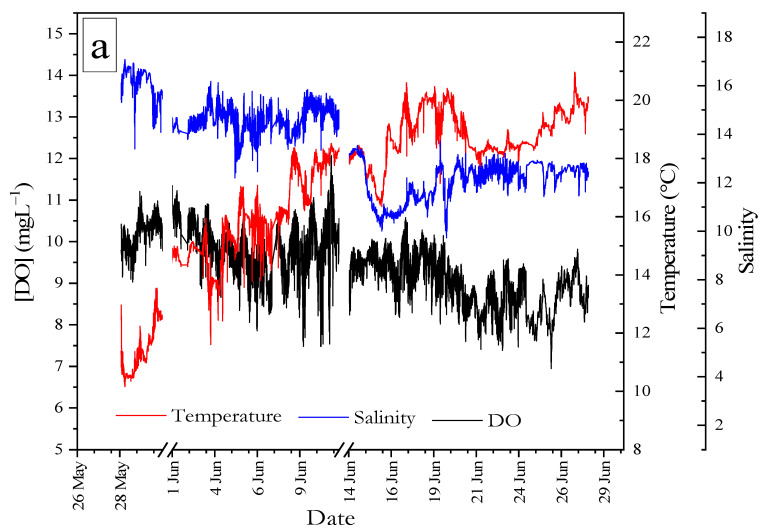
(**a**) Thirty-day time series (28 May to 28 June 2021) of the environmental parameters at Kiel Fjord including dissolved oxygen (DO) (black), salinity (blue), and temperature (red) at a 1 min sampling frequency (*n* = 32,820) recorded by the YSI sensor deployed near the AutoLAb analyzer intake. (**b**) Time series data from the period 12 May to 26 June 2021, for wind speed (blue lines, left Y-axis) and water temperature (red lines, right Y-axis) obtained from GEOMAR weather metrological station. (**c**) Time series data from the period 28 May to 27 June 2021 from CONTROS HydroC-CO_2_ for pCO_2_ data.

**Figure 8 sensors-22-03479-f008:**
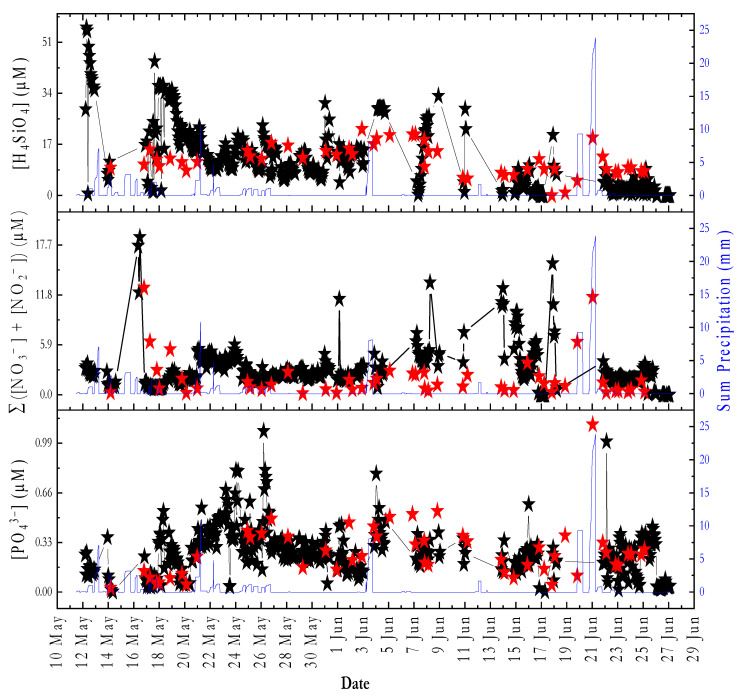
Time series data for the period of 12 May to 27 June 2021 of *on-site* PO_4_^3−^, Σ(NO_3_^−^ + NO_2_^−^), and H_4_SiO_4_ analyzer measurements (black stars) obtained from an *on-site* analyzer and from discrete samples analyzed using a laboratory-based segmented flow analyzer (red stars). Sum precipitation (i.e., rainfall data) were shown as blue lines. The nutrient concentrations were calculated by applying linear regression using four onboard standards.

**Figure 9 sensors-22-03479-f009:**
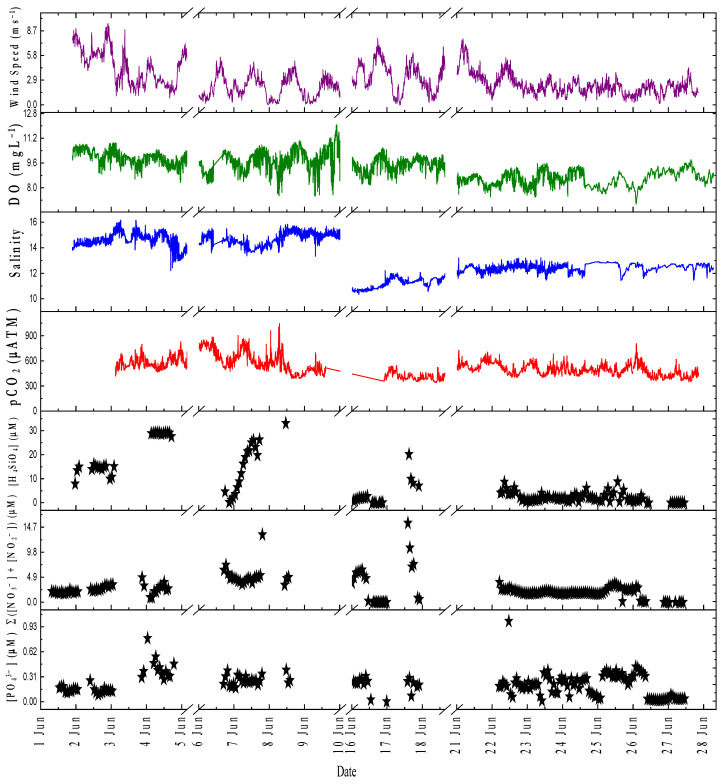
Time series data for the period from 2 June to 10 June 2021 and from 17 June to 27 June. June 2021 for PO_4_^3−^, Σ(NO_3_^−^ + NO_2_^−^), and H_4_SiO_4_ in µM represented by black stars; pCO_2_ in µatm (red lines), DO in mgL^−1^ (green lines), salinity (blue lines), and wind speed in ms^−1^ (purple lines).

## Data Availability

Data is contained within the article and the Appendix A.

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
