# Peer review of "Improvement of On-Site Sensor for Simultaneous Determination of Phosphate, Silicic Acid, Nitrate plus Nitrite in Seawater"

_sensors, 2022, doi:10.3390/s22093479_

Round 1

Reviewer 1 Report

The manuscript by Altahan et al. describes their procedure to improve the detection limits of a commercial automated sensor for the detection of macronutrients (phosphate (PO43−), nitrate (NO3−), and silicic acid (H4SiO4)) in seawater by optimizing certain key parameters important for analysis. Although the technique described is very interesting and the application for improved nutrient detection in water samples, especially seawater, is extremely critical for the environmental monitoring of water to ensure its safety and has been seeing a lot of work from different research groups across the globe, the report as it currently stands requires a major revision before acceptance. Below are some of the comments I have on what is presented in the manuscript:

The manuscript should be properly revised to improve the grammar being used. The overall quality is acceptable but there are some mistakes that can be avoided (e.g. “in southwestern Baltic Sea”, “with a water supply from 1 m depth”, “are labour intensive and expensive and typically”, line 627, etc…)

In the abstract, the authors mention that “analyte to reagent ratio for NO3− analysis” was optimized. What about the analyte to reagent ratio for nitrite analysis? Since nitrite is mentioned in the title of the manuscript.

All of the limits of detection (LOD) calculated fall just a bit short (0.02-0.05 μM) of the concentrations being tested.  Mako et al. make the claim that results would be considered more accurate if the LOD falls within the concentration range being tested. How would the authors respond to this claim to justify the accuracy of their LOD results? 

  1. Mako, T.L.; Levenson, A.M.; Levine, M. Ultrasensitive Detection of Nitrite through Implementation of N-(1-Naphthyl)ethylenediamine-Grafted Cellulose into a Paper-Based Device. ACS Sens. 2020, 5, 1207–1215.

Limit of quantification (LOQ) should be included and not just the limit of detection. There are several ways to calculate the LOQ. The authors are referred to the below article by Belter et al. for a list of different methods that can be used to calculate and report the LOQ. The authors can also use plus (or minus) 10 times the standard deviation to the mean of the blank to calculate the LOQ.

  1. Belter, M.; Sajnóg, A.; Barałkiewicz, D. Over a century of detection and quantification capabilities in analytical chemistry—Historical overview and trends. Talanta 2014, 129, 606–616.

The authors provided the Relative Standard Deviation (RSD) for phosphate, silicic acid, and nitrate but not that for nitrite. They would have to add the RSD for nitrite as well since nitrite is in the title of the article.

What criterion did the authors use to deduce that the analyzer showed good precision? Was there a critical RSD threshold value they didn’t exceed or were there other criteria as well?

In the introduction section, the authors mention the different techniques such as optical, electrochemical and wet chemical techniques that are currently being employed for on-site monitoring of nutrients in marine waters. However, they failed to mention a new technique which has been garnering a lot of research interest and attention in the past few years and that is based on the use of paper-based devices for on-site monitoring for nutrients in seawater. These systems are relatively inexpensive compared to traditional LOC technology, exhibit good detection limits, are easy to operate by untrained users, not affected by biofouling, can be easily deployed for point of care diagnostics and then easily disposed of after use. One drawback is they don’t allow for continuous monitoring similar to automated sensors. The authors are referred to the following publications for a quick reference:

  1. Charbaji, H. Heidari-Bafroui, C. Anagnostopoulos, M. Faghri, A New Paper-Based Microfluidic Device for Improved Detection of Nitrate in Water, Sensors. 21 (2021). https://doi.org/10.3390/s21010102
  2. M. Racicot, T.L. Mako, A. Olivelli, M. Levine, A Paper-Based Device for Ultrasensitive, Colorimetric Phosphate Detection in Seawater, Sensors. 20 (2020). https://doi.org/10.3390/s20102766

In line 110-111, the authors make the claim “However, the protocols show a poor performance and precision, which limits their use for on-site environmental applications” without providing any detail or analysis on how they came to this conclusion.

In line 124-125, the authors have to clarify the chemical being used in “soaked in 1 M (37 %, Carl Roth, Karlsruhe, Germany)”

In line 147, the authors have to clarify the concentration of HCL being used in “Then, 15 ml of concentrated HCl (trace-metal grade, Fisher Scientific, 147 Waltham, Massachusetts, US) was added”.

In lines 162-164, the authors have to clarify the storage conditions (room conditions, field conditions, fridge, freezer, etc.) of the reagents and for how long before use.

The quality of figure 1 can be improved by reorganizing the labels, making sure they are all of the same font, checking the grammar and making sure the labels are similar to what is mentioned in the caption.

The authors mention the peak absorbance (880 nm for phosphate and silicic acid and 540 nm for nitrate and nitrite measurements) of the reactions involved but don’t provide the peak wavelength or the range of the light source being used since the silicic acid and nitrate and nitrite use the same detector chamber. The range of wavelength being used affects detection limits and the authors are referred to the following:

  1. Heidari-Bafroui, A. Charbaji, C. Anagnostopoulos, M. Faghri, A Colorimetric Dip Strip Assay for Detection of Low Concentrations of Phosphate in Seawater, Sensors. 21 (2021). https://doi.org/10.3390/s21093125

The supplementary document folder didn’t include Video S1.

Lines 278-281 have to be rewritten to clarify the concentration of the standard solution used for each nutrient of interest.

There seems to be a typo in line 307, authors are encouraged to double check if they meant the value of 0.0173 instead of 0.173 in “for 0 s to 0.173 (± 0.0002) AU μM−1 and 1.19% RSD for 90 s”. This concentration at time 90 second is not present in Figure 4a.

In figure 4 (a) and (b), the increment on the x-axis should be changed from 25 to 30 since the times tested are a multiple of 30 and not 25. This will make it easier to see the points tested on the graph. Also, the labels on the x-axis and y-axis should be modified and the units placed within brackets and not after the division sign to reduce confusion. The Sensitivity label on the y-axis should be modified to include the units in parenthesis and not after the division sign.

The figure for reduction efficiency as a function of reduction time has to be included either in the manuscript or the supplementary information document since this is one of the key parameters that was optimized and has an effect on reaction kinetics. The authors should mention the RSD value with the reduction efficiency. Also, the authors should clearly mention how the reduction efficiency was calculated. The authors mentioned the disadvantages of using a cadmium column, but did they consider using zinc microparticles?

What’s the difference between the “reduction efficiency” mentioned in text and the reduction ratio used in figure 4c. Authors are encouraged to use one term and to include the RSD values in text.

Lines 360 to 363 need to be rewritten to make it easier to read and clearer to comprehend when looking at the graph in figure 5a. Also, the increment on the x-axis of all graphs in Figure 5 should be changed from 5 to 7. Since the authors have concentrations that are multiples of 7 for the most part aside from 23.

In the caption of Figure 5 and 6, the Error bar (±1 β) whereas in Figure 4, the Error bar (±1 SD). Is β the same as standard deviation or is it calculated differently? Please clarify in the text or be consistent throughout the text for figures in the manuscript and for figures in the supplementary document where some are missing this information and have empty brackets () in the caption.

Line 367 should be modified to “H4SiO4 in deionized water S = 0 (0.07 AU)” to be consistent with the rest of the text.

Between lines 373 and 375, the authors claim “ This is evident when comparing the analytical sensitivity of 1 μM PO43− standard in a solution of deionized water (S = 0) (0.008 AU) with that using a solution of S = 7 (0.008 AU)”; however, this data is not included in figure 5c that has absorbance for 0.5 μM PO43− standard and it’s also not included in the supplementary file.

Lines 382-390 need to be rewritten to make it clear that the optical correction employed by the authors in equation 1 decreases the underestimation of concentration using the traditional Beer's law equation.

A note to the authors, the reagents used in the Griess assay decay and change color with respect to time, especially the sulfanilamide. Therefore, it’s critical to have proper storage and run all experiments on salinity on the same day to make sure that reagent stability is not a factor in the results. Figure S1a, shows an underestimation close to %50 with a salinity of 14. Therefore, it’s crucial to run a proper design of experiment to limit the effect of all other factors, one of them being reagent shelf life. The use of vanadium III chloride creates the “dark-turquoise” color that makes it difficult to “notice” the change in color of the colorless Griess assay with time. Did the authors look at the absorbance value for a constant concentration in DI water as a function of time for the different reagents used at a temperature similar to the one encountered in the field?

  1. Hakobyan L, Monforte-Gómez B, Moliner-Martínez Y, Molins-Legua C, Campíns-Falcó P. Improving Sustainability of the Griess Reaction by Reagent Stabilization on PDMS Membranes and ZnNPs as Reductor of Nitrates: Application to Different Water Samples. Polymers. 2022; 14(3):464. https://doi.org/10.3390/polym1403046

In line 401, R2 is called coefficient of determination and not detection.

In lines 408-410, how does the concentrations tested, or calibration graphs created play any role in the limit of detection calculated? Limit of quantification values should also be mentioned.

The font in lines 482- 488 and 498-504 is different than the rest of the manuscript.

Why are there 2 gaps in figure 7a? Was the sensor also being serviced?

Lines 524-523 need to be rewritten to make it clearer how the rainfall data was monitored and at what times. Twice vs. three times a day?

In lines 543-545, the p-value for the paired t-test for the Σ (NO3- + NO2-) was 0.04 which shows that there’s a significant difference in the means if a value is taken as the customary 0.05.

Finally, the authors are encouraged to mention biofouling and how they expect to guard against it during field deployment. Also, how they expect to ensure that the copper net doesn’t get blocked by large particles or sediments again as this has previously affected their results. 

Author Response

Response to reviewers

Dear Editor,

Many thanks for handling our manuscript. We like to thank the reviewers for their inspiring words, and very much appreciate the comments by the reviewers, and have followed up their advice and amended the manuscript accordingly. Below, we present the reviewers’ comments with our replies. We hope that we have satisfied the requirements by you and the reviewers.  

Below, you will find responses to reviewers’ comments for our article entitled

Improvement of on-site sensor for simultaneous determination of phosphate, silicic acid, nitrate plus nitrite in seawater

Comments:

  1. The manuscript should be properly revised to improve the grammar being used. The overall quality is acceptable but there are some mistakes that can be avoided (e.g. “in southwestern Baltic Sea”, “with a water supply from 1 m depth”, “are labour intensive and expensive and typically”, line 627, etc…)

Done. The whole manuscript was revised

  1. In the abstract, the authors mention that “analyte to reagent ratio for NO3− analysis” was optimized. What about the analyte to reagent ratio for nitrite analysis? Since nitrite is mentioned in the title of the manuscript.

 First, nitrite concentrations in natural waters are very low compared to nitrate concentrations. Nitrate concentrations in surface waters range from nanomolar in oligotrophic regions to several hundred micromolar in nutrient-rich rivers. Nitrite concentrations are typically two orders of magnitude lower than nitrate concentrations in oxic water. We therefore chose not to determine nitrite separately from nitrate using our anaylzer and the following statement was added to the manuscript “Technically, the analyzer can detect nitrite in the field, as we described in Materials and Methods. However, nitrite concentrations in natural waters are typically in the nanomolar range and below our reported LOD, which limits the use of our analyzer to detect nitrite and nitrate separately.”, and also the title of the manuscript was changed as follows “Improvement of on-site sensor for simultaneous determination of phosphate, silicic acid, nitrate plus nitrite in seawater.”

  1. All of the limits of detection (LOD) calculated fall just a bit short (0.02-0.05 μM) of the concentrations being tested. Mako et al. make the claim that results would be considered more accurate if the LOD falls within the concentration range being tested. How would the authors respond to this claim to justify the accuracy of their LOD results?
  2. Mako, T.L.; Levenson, A.M.; Levine, M. Ultrasensitive Detection of Nitrite through Implementation of N-(1-Naphthyl)ethylenediamine-Grafted Cellulose into a Paper-Based Device. ACS Sens. 2020, 5, 1207–1215.

The limit of detection (LOD) and limit of quantification (LOQ) are used to evaluate the data quality of analysed samples. LOD is typically used to decide whether the analyte is present, i.e., the true concentration that is greater than zero, while LOQ is used to decide whether the concentration of an analyte can be quantified or determined. Therefore, LOD may be slightly below the lowest concentration value determined during calibration, while LOQ chould be within the concentration range, as shown in the following table. If LOD is within the concentration range, it means that LOQ is far from the lowest concentration determined in the calibration curve, which is not consistent with the definition of LOQ.

An appropriate way of assessing accuracy of measurements is by measurement of certified reference materials with concentrations relevant to the samples. This approach was conducted and showed our measurements are accurate.

LOD (µM)

LOQ (µM)

Concentration range

(µM)

NO3-

0.45

1.5

0.5 - 100

NO2-

0.27

0.89

0.4 – 100

PO43-

0.18

0.6

0.2 – 100

H4SiO4

0.15

0.3

0.2 - 100

The definition of LOD and LOQ, reported and reviewed by Zorn et al. is clear.

Zorn, M. E., Gibbons, R. D., & Sonzogni, W. C. (1999). Evaluation of approximate methods for calculating the limit of detection and limit of quantification. Environmental science & technology, 33(13), 2291-2295.

  1. Limit of quantification (LOQ) should be included and not just the limit of detection. There are several ways to calculate the LOQ. The authors are referred to the below article by Belter et al. for a list of different methods that can be used to calculate and report the LOQ. The authors can also use plus (or minus) 10 times the standard deviation to the mean of the blank to calculate the LOQ.
  2. Belter, M.; Sajnóg, A.; Barałkiewicz, D. Over a century of detection and quantification capabilities in analytical chemistry—Historical overview and trends. Talanta 2014, 129, 606–616.

Done. LOQ was calculated like LOD was calculated according to the IUPAC recommendation. Equations 3 and 4 were added to the manuscript with the detailed description of the term used, and the relevant paragraph was rewritten as follow “The limit of detection (LOD) was calculated as 0.18 μM, 0.15 μM, 0.45 μM, and 0.269 μM and the limit of quantification (LOQ) was calculated as follow 0.6 µM 0.3 µM, 1.5 µM  and 0.89 µM for PO43−, H4SiO4, NO3, and NO2, respectively, where LOD and LOQ were calculated according to IUPAC recommendation using the following equations:

(3)

 (4)

Where σ is defined as the standard deviation of blank measurements (n=10).”, and reference 51 was cited.

  1. The authors provided the Relative Standard Deviation (RSD) for phosphate, silicic acid, and nitrate but not that for nitrite. They would have to add the RSD for nitrite as well since nitrite is in the title of the article.

The RSD we reported is for NO3- plus NO2- (Σ (NO3 + NO2)), as it is not possible to reliably detect NO2- separately from NO3-. The certified concentration of NO2- in Kanso CRM CG (0.06 µM) is lower than our reported LOD for nitrate. The label in Figure S2 is also corrected to Σ (NO3 + NO2).

  1. What criterion did the authors use to deduce that the analyzer showed good precision? Was there a critical RSD threshold value they didn’t exceed or were there other criteria as well?

The criteria used were based on the suggestions of Gibbons et al. who suggested that an RSD value of less than 10% was acceptable.

Gibbons, R. D., & Coleman, D. E. (2001). Statistical methods for detection and quantification of environmental contamination. The following sentence was added “The values of RSD are less than the extent reported by Gibbons et al. (10 % RSD) showed good precision of the analyzer.”, and reference 70 was cited.

  1. In the introduction section, the authors mention the different techniques such as optical, electrochemical and wet chemical techniques that are currently being employed for on-site monitoring of nutrients in marine waters. However, they failed to mention a new technique which has been garnering a lot of research interest and attention in the past few years and that is based on the use of paper-based devices for on-site monitoring for nutrients in seawater. These systems are relatively inexpensive compared to traditional LOC technology, exhibit good detection limits, are easy to operate by untrained users, not affected by biofouling, can be easily deployed for point of care diagnostics and then easily disposed of after use. One drawback is they don’t allow for continuous monitoring similar to automated sensors. The authors are referred to the following publications for a quick reference:
  2. A. Charbaji, H. Heidari-Bafroui, C. Anagnostopoulos, M. Faghri, A New Paper-Based Microfluidic Device for Improved Detection of Nitrate in Water, Sensors. 21 (2021). https://doi.org/10.3390/s21010102
  3. J.M. Racicot, T.L. Mako, A. Olivelli, M. Levine, A Paper-Based Device for Ultrasensitive, Colorimetric Phosphate Detection in Seawater, Sensors. 20 (2020). https://doi.org/10.3390/s20102766

Done. The following paragraph was added to the introduction “Recently, new paper-based microfluidic devices for the determination of macronutrients in natural waters have been reported. The techniques are based on fluid flow through paper by capillary action without the need for a pump. In principle, the device consists of a sample port into which the water sample is introduced and transport channels connecting other parts of the device, such as the reaction zone, where the analyte solution mixes or reacts with the reagents. The signal (i.e., color formation) is subsequently formed in the detection zone, and can be quantified using a cell phone or desktop scanner. Although the proposed systems offer promising applications for on-site observations of nutrients in natural waters, the technique does not allow autonomous continuous monitoring.”, and references 27 and 28 were cited.

  1. In line 110-111, the authors make the claim “However, the protocols show a poor performance and precision, which limits their use for on-site environmental applications” without providing any detail or analysis on how they came to this conclusion.

Done. The following paragraph was addedIn the stored protocol, only one standard was used for each nutrient. There is no matrix effect correction (i.e., no optical correction) in the sample concentration calculation, as described in the Data Processing Protocol section of the User's Guide, which limits the use of the analyzer in field deployments. The conventional cadmium column reduction procedure for nitrate determination, which requires regular regeneration, and the rate at which reagents and standards are consumed per measurement, also limit its use for long-term field use.

  1. In line 124-125, the authors have to clarify the chemical being used in “soaked in 1 M (37 %, Carl Roth, Karlsruhe, Germany)”.

Done. The following statement was corrected as follow “soaked in 1 M HCl (37 %, Carl Roth, Karlsruhe, Germany)”

  1. In line 147, the authors have to clarify the concentration of HCL being used in “Then, 15 ml of concentrated HCl (trace-metal grade, Fisher Scientific, 147 Waltham, Massachusetts, US) was added”.

Done. The statement was corrected as follow “Then, 15 ml of concentrated HCl (37 %, trace-metal grade, Fisher Scientific, Waltham, Massachusetts, US) was added.”

  1. In lines 162-164, the authors have to clarify the storage conditions (room conditions, field conditions, fridge, freezer, etc.) of the reagents and for how long before use.

Done. The following statement was rewritten as follow “All reagent solutions were stored in brown 500 ml high-density polyethylene (HDPE) laboratory-grade bottles (Nalgene, Thermo Scientific, US) and kept refrigerated when not in use. Blank, standard, and cleaning solutions were freshly prepared prior to field use and stored in 1000 ml HDPE Nalgene bottles.”

  1. The quality of figure 1 can be improved by reorganizing the labels, making sure they are all of the same font, checking the grammar and making sure the labels are similar to what is mentioned in the caption.

Done. All the labels are now of the same font (font 19, Garamond) and are corrected to be similar as the caption.

  1. The authors mention the peak absorbance (880 nm for phosphate and silicic acid and 540 nm for nitrate and nitrite measurements) of the reactions involved but don’t provide the peak wavelength or the range of the light source being used since the silicic acid and nitrate and nitrite use the same detector chamber. The range of wavelength being used affects detection limits and the authors are referred to the following:
  2. H. Heidari-Bafroui, A. Charbaji, C. Anagnostopoulos, M. Faghri, A Colorimetric Dip Strip Assay for Detection of Low Concentrations of Phosphate in Seawater, Sensors. 21 (2021). https://doi.org/10.3390/s21093125

Silicic acid, nitrate and nitrite detectors have the same housing (chamber), but they have two separate colorimeters with two different flow cells and two separate photodiodes and LEDs. We mentioned this in the manuscript where we described the components of the AutoLAB, which consists of 3 colorimetric detectors.
And for the wavelengths of the LEDs and photodiodes used, the following statement was added as follows “A
green LED with a peak wavelength of 567 nm and a silicon photodiode with a peak intensity at a wavelength of 570 nm was used for detection of Σ(NO3- + NO2-). No information on LEDs or photodiodes for the detectors of PO43- or H4SiO4 is provided in the operating manual.

  1. The supplementary document folder didn’t include Video S1.

Done. Video S1 was uploaded onto Figshare with the following doi 10.6084/m9.figshare. 19608597.v1.

  1. Lines 278-281 have to be rewritten to clarify the concentration of the standard solution used for each nutrient of interest.

Done. The statement was rewritten as follow “The analyzer was equipped with a blank solution and three standard solutions for NO3- (1, 5, and 10 μM), PO43- (0.5, 1, and 2 μM), and H4SiO4 (1, 10, and 20 μM), all prepared in artificial seawater (17 g L-1 NaCl).”

  1. There seems to be a typo in line 307, authors are encouraged to double check if they meant the value of 0.0173 instead of 0.173 in “for 0 s to 0.173 (± 0.0002) AU μM−1 and 1.19% RSD for 90 s”. This concentration at time 90 second is not present in Figure 4a.

Done. It is corrected and the statement was corrected as follow “the analytical sensitivity showed increases of 0.0058 (± 0.0025) AU µM−1 and an RSD of 43.9% for 0 s to 0.173 (± 0.0002) AU µM−1 and 1.19% RSD for 120 s.”

  1. In figure 4 (a) and (b), the increment on the x-axis should be changed from 25 to 30 since the times tested are a multiple of 30 and not 25. This will make it easier to see the points tested on the graph. Also, the labels on the x-axis and y-axis should be modified and the units placed within brackets and not after the division sign to reduce confusion. The Sensitivity label on the y-axis should be modified to include the units in parenthesis and not after the division sign.

Done. All the modifications were performed and applied to all figures in the manuscript.

  1. The figure for reduction efficiency as a function of reduction time has to be included either in the manuscript or the supplementary information document since this is one of the key parameters that was optimized and has an effect on reaction kinetics. The authors should mention the RSD value with the reduction efficiency. Also, the authors should clearly mention how the reduction efficiency was calculated. The authors mentioned the disadvantages of using a cadmium column, but did they consider using zinc microparticles?

Done. Figure S1 was added to show the influence of the reaction time on the reduction efficiency and absorbance of a definite concentration of nitrate. The added caption reads as follow “Effect of reaction time in minutes on (a) the absorbance of 10 µM NO3- and (b) the reduction efficiency (%) which is defined as the ratio of the absorbance of 10 µM NO3- and the absorbance of 10 µM NO2-.” Considering the potential use of zinc microparticles as reductant the following sentence was added to the introduction “It showed a more promising performance for long-term use than the classic copper-coated cadmium column or zinc microparticles reported by Ellis et al. in 2011, which must be replaced daily due to degradation.”, and reference 39 was cited.

  1. What’s the difference between the “reduction efficiency” mentioned in text and the reduction ratio used in figure 4c. Authors are encouraged to use one term and to include the RSD values in text.

Done. The label of the right y-axis in Figure 4, C was corrected to show reduction efficiency instead of reduction ratio.

  1. Lines 360 to 363 need to be rewritten to make it easier to read and clearer to comprehend when looking at the graph in figure 5a. Also, the increment on the x-axis of all graphs in Figure 5 should be changed from 5 to 7. Since the authors have concentrations that are multiples of 7 for the most part aside from 23.

Done. The statement was corrected as follow “Figure 5a shows an absorbance of 5 µM NO3- in deionized water S = 0 (0.14 AU), with absorbance values decreasing with increasing salinity from S = 7 (0.105 AU) to S = 14 (0.09 AU). A steady state condition was reached with increasing salinity to S = 23 (0.09 AU) and to S = 35 (0.09 AU)” and the increment in figure 5a was changed.

  1. In the caption of Figure 5 and 6, the Error bar (±1 β) whereas in Figure 4, the Error bar (±1 SD). Is β the same as standard deviation or is it calculated differently? Please clarify in the text or be consistent throughout the text for figures in the manuscript and for figures in the supplementary document where some are missing this information and have empty brackets () in the caption.

Done. It was corrected as follow “Error bar (±1 SD)” in Figure 5 and 6 and was added in Figure S1. And the symbols were added in the brackets.

  1. Line 367 should be modified to “H4SiO4 in deionized water S = 0 (0.07 AU)” to be consistent with the rest of the text.

Done.

  1. Between lines 373 and 375, the authors claim “This is evident when comparing the analytical sensitivity of 1 μM PO43− standard in a solution of deionized water (S = 0) (0.008 AU) with that using a solution of S = 7 (0.008 AU)”; however, this data is not included in figure 5c that has absorbance for 5 μM PO43− standard and it’s also not included in the supplementary file.

Done. The concentration of PO43− was corrected in the text to be “This is evident when comparing the analytical sensitivity of 0.5 µM PO43− standard in a solution of deionized water”.

  1. Lines 382-390 need to be rewritten to make it clear that the optical correction employed by the authors in equation 1 decreases the underestimation of concentration using the traditional Beer's law equation.

Done. The paragraph was rewritten as follow “We corrected for salinity errors during measurements by taking the photodiode measurements for the analyte solution before adding the reagents. Figure S1 shows the measured concentrations of 5 µM NO3- (Figure S1, a), 5 µM H4SiO4 (Figure S1, b), and 1 µM PO43- (Figure S1, c). The values obtained with the traditional Beer's law equation are shown as red circles, while those obtained with equation 1 using the optical correction are shown as black circles. The comparison between the two values indicates that values obtained with the traditional Beer's law were underestimated by 3.95 % (S = 0) compared to values obtained following the optical correction. The underestimation increased with increasing salinity to 40.6 % (S = 23 & S = 35) for 5 µM NO3-, while an underestimation of 2.5 % (S = 0) was found for 5 µM H4SiO4, increasing to 43.9 % (S = 35) with increasing salinity, and an underestimation of 1.42 % (S = 0) was found for 1 µM PO43-, increasing to 16.4 % (S = 35) with increasing salinity. Despite the usefulness of the optical correction, it is recommended to use standards with salinity close to that of the studied waters for field work activities.

  1. A note to the authors, the reagents used in the Griess assay decay and change color with respect to time, especially the sulfanilamide. Therefore, it’s critical to have proper storage and run all experiments on salinity on the same day to make sure that reagent stability is not a factor in the results. Figure S1a, shows an underestimation close to %50 with a salinity of 14. Therefore, it’s crucial to run a proper design of experiment to limit the effect of all other factors, one of them being reagent shelf life. The use of vanadium III chloride creates the “dark-turquoise” color that makes it difficult to “notice” the change in color of the colorless Griess assay with time. Did the authors look at the absorbance value for a constant concentration in DI water as a function of time for the different reagents used at a temperature similar to the one encountered in the field?
  2. Hakobyan L, Monforte-Gómez B, Moliner-Martínez Y, Molins-Legua C, Campíns-Falcó P. Improving Sustainability of the Griess Reaction by Reagent Stabilization on PDMS Membranes and ZnNPs as Reductor of Nitrates: Application to Different Water Samples. Polymers. 2022; 14(3):464. https://doi.org/10.3390/polym1403046

We did not conduct reagent stability experiments s part of this study. The experiments on the effect of salinity on the nitrate signal were conducted on the same day and therefore reagent decay can be excluded as a cause for the changes. We studied the effects of salinity to determine the importance of the optical correction for the calculation of absorbance implemented for the data processing of the output of the analyzer, and to highlight the importance of preparing the standards with the same salinity as the water studied. Other parameters such as the stability of the reagent with temperature, are of low relevance to the field work measurements, since the calibration curve is typically performed daily under the same environmental conditions, compensating for changes such as the shelf life of the reagent or temperature effects.

  1. In line 401, R2 is called coefficient of determination and not detection.

Done. Corrected to read “with a coefficient of determination R2”

  1. In lines 408-410, how does the concentrations tested, or calibration graphs created play any role in the limit of detection calculated? Limit of quantification values should also be mentioned.

Done. The same answer to question 4 where we added the description of the term σ in equations 3 and 4, where the calibration curve is included to calculate σ, which is the standard deviation of the measurements made for the blank signals. The limit of quantification (LOQ) was calculated and mentioned in the text.

  1. The font in lines 482- 488 and 498-504 is different than the rest of the manuscript.

Done. The font is corrected to be the same as the whole manuscript.

  1. Why are there 2 gaps in figure 7a? Was the sensor also being serviced?

The sensor was serviced and there was a problem in downloading the data and the following statement was added "with two gaps on June 12 and June 11-14 due to a problem in downloading the data from the sensor."

  1. Lines 524-523 need to be rewritten to make it clearer how the rainfall data was monitored and at what times. Twice vs. three times a day?

Done and the statement was rewritten as follow “Rainfall data was monitored as the sum of the precipitation over a period of 12 hours, and measured three times per a day at 0, 6 and 18 h UTC (Figure 8, blue lines).”

  1. In lines 543-545, the p-value for the paired t-test for the Σ (NO3- + NO2-) was 0.04 which shows that there’s a significant difference in the means if  value is taken as the customary 0.05.

All the statistical relations were tested at a significance level of 1 % and the following statement was corrected as follow “there is no significant difference between the means at 1 % level”

  1. Finally, the authors are encouraged to mention biofouling and how they expect to guard against it during field deployment. Also, how they expect to ensure that the copper net doesn’t get blocked by large particles or sediments again as this has previously affected their results.

Done. The following paragraph was added One way to avoid fouling of internal analyzer components problem is to use a tubing with a narrow inner diameter and a slow flow rate (same approach is used for Sunburst devices). This was not possible with the aquarium pump used in our study as it was not possible to control the flow rate. In future applications we will use a small pore size (e.g. 1 μm) syringe filter which will prevent internal fouling.”, and reference 76 was cited.

Reviewer 2 Report

The paper deals with improvement of on-site sensor based on FIA technology for simultaneous determination of phosphate, silicic acid and nitrate/nitrite in seawater. A batch of experimental data together with results is presented; however, there are some doubts and remarks:

  1. The determination of phosphate and silicate is based on the similar chemical reaction of polyoxometalate formation. There are many papers describing the simultaneous kinetic analysis of both species – see papers of S.R. Crouch in ref. [39] therefore the attention of several experimental conditions (e.g., reaction time, acidity, etc.) should be taken and optimized.
  2. What is difference between reaction time and complexation time (e.g., abstract, Fig. 4A vs. 4B)?
  3. The sensor is based on FIA method – some relevant infos (e.g., books, papers) of pioneers (E. Hanssen, J. Ruzicka] are missing. In addition, FIALab company is producing equipment’s for application of online analysis of sea components – see https://www.flowinjection.com/ or

https://www.flowinjectiontutorial.com/index.html

  1. Video S1 is not available in ESI.
  2. The description of FIA equipment should be in more detail since it is not available elsewhere. In addition, AUTOLAB is equipment from Netherland is also employed in voltametric analysis……
  3. [29] is wrong because of missing pages. Did authors read this old paper written in German language or not?
  4. The standard deviations of parameters (e.g., sensitivity, intercept) for calibration plots (Fig. 6A, 6B, 6C and 6D) are missing in order to use the Student t-test about the significance of intercept since it is assumed that this value of intercept should be zero….. Or not as the blank contribution? This is also important for LOD estimation.  
  5. The analysis of CRM’s (lines 452-459) has not been evaluated by the Student t-test to get information about the systematic error, i.e. bias, of the analytical procedure. In addition, the standard-addition method using the sample of Kiel-sea sample should be applied to get information about the recovery and thus, the effect of sample salinity could be eliminated.
  6. It is not clear what is the novelty of paper since the basic parameters – figures of merit (see Table 1) before optimization is missing. In addition, the LOD’s are almost the same for other FIA analyzers (see table 1).  
  7. The careful interpretation of experimental data for phosphate (mean value 0.26 mikroM] should be taken since LOD is about 0.18 mikroM.

I could not recommend this paper to publish in SENSORS journal in this stage because the novelty of this paper is not clear. In addition, the analytical procedure should be optimized to be robust for analysis of seawater samples of different origin using standard-addition method.   

Author Response

Response to reviewers

Dear Editor,

Many thanks for handling our manuscript. We like to thank the reviewers for their inspiring words, and very much appreciate the comments by the reviewers, and have followed up their advice and amended the manuscript accordingly. Below, we present the reviewers’ comments with our replies. We hope that we have satisfied the requirements by you and the reviewers.  

Below, you will find responses to reviewers’ comments for our article entitled

Improvement of on-site sensor for simultaneous determination of phosphate, silicic acid, nitrate plus nitrite in seawater

Comments:

  1. The determination of phosphate and silicate is based on the similar chemical reaction of polyoxometalate formation. There are many papers describing the simultaneous kinetic analysis of both species – see papers of S.R. Crouch in ref. [39] therefore the attention of several experimental conditions (e.g., reaction time, acidity, etc.) should be taken and optimized.

We appreciate the comment by the reviewer. The influence of proton concentrations on the determination of silicic acid and phosphate has however been thoroughly studied and reported in the literature. A commonly employed (H+/MoO4-2) ratio is 60 - 80, which can be achieved at a pH of 0.4 - 0.9. The use of potassium antimony tartrate for phosphate determination and oxalic acid as a masking agent in silicic acid determination are absolutely necessary to eliminate the mutually interfering effects. Therefore, we have chosen not to investigate these variables again for our analyzer set-up.

  1. What is difference between reaction time and complexation time (e.g., abstract, Fig. 4A vs. 4B)?

Both terms are identified in section 3.1. We would like to distinguish between them because the reaction time was used for phosphate, where all reagents (mixed molybdate reagent (i.e., ammonium molybdate solution and potassium antimony tartrate as masking agent) and the reducing agent (i.e., ascorbic acid) are added once to the analyte solution and then undergo reaction during the waiting time. For the complexation time, only the molybdate reagent was added to the analyte solution and the complexation was run for a specified time until the yellow silicomolybdate complex was formed prior to the addition of the other two reagents (oxalic acid and ascorbic acid) after which a further waiting time was applied. While the latter waiting time had no effect on analytical sensitivity, this was not true for the first case. Therefore, the term complexation time was used to distinguish between the two waiting times. 

  1. The sensor is based on FIA method – some relevant infos (e.g., books, papers) of pioneers (E. Hanssen, J. Ruzicka] are missing. In addition, FIALab company is producing equipment’s for application of online analysis of sea components – see https://www.flowinjection.com/ or https://www.flowinjectiontutorial.com/index.html

Done and references 30 and 31 were added.

  1. Video S1 is not available in ESI.

Apologies. Done and Video S1 was uploaded into Figshare with the following doi 10.6084/m9.figshare. 19608597.v1.

  1. The description of FIA equipment should be in more detail since it is not available elsewhere. In addition, AUTOLAB is equipment from Netherland is also employed in voltametric analysis……

Done. The following paragraph was added “To minimize light interference from the outside, the colorimeters were encapsulated in polyurethane. Inside the electronics housing is a series of electronic modules: the main control unit and the motor drivers and detector interfaces. Both the motor drivers and detector interfaces have their own microprocessors and are controlled by the main control unit via a link. Four devices (syringe motor, valve motor, phosphate detector, and ((nitrate + nitrite) and silicic acid detectors) are configured through an arrangement called a serial peripheral system (SPS), where the detectors and motors are referred to as SPS devices and each device has its own SPS address, which is called in the internal scripting language.

AutoLAB is the commercial name we got from the operating manual from EnironTech LLC for the automatic nutrient analysis system. The AUTOLAB systems from the Netherlands are electrochemical systems (Ecochemie/Metrohm), and not related to the analyzer mentioned in this paper.

  1. [29] is wrong because of missing pages. Did authors read this old paper written in German language or not?

Reference 29 was the citation of the first reported work for the Griess method by Griess et al. in 1879, which is commonly cited in various published reports on the spectrophotometric determination of nitrite and hence nitrate.

  1. The standard deviations of parameters (e.g., sensitivity, intercept) for calibration plots (Fig. 6A, 6B, 6C and 6D) are missing in order to use the Student t-test about the significance of intercept since it is assumed that this value of intercept should be zero….. Or not as the blank contribution? This is also important for LOD estimation.

Many thanks for the comment. Done. Table S1 was added, which contains all the values for the intercept and slope, as well as the standard deviations, t values, and probability values, and the following statement was added to the manuscript: " The values of the intercepts and slopes of the corresponding calibration curves are presented in Table S1, standard deviation values, static t-values and probabilities are also reported, and the data showed that all values were significant (p < 0.01), except for the intercept of the calibration curve for silicic acid, because the blank measurements for silicic acid in deionized water showed negative values.", and reference 49 was added.

The values of the axis intercept can only be zero if we take the blank response into account when calculating absorbance, since the blank itself has an absorbance value that comes from either the matrix or the reagent itself, which could be colored. However, using the equation to calculate absorbance with or without blank correction does not matter in the measurements and, of course, does not affect the estimate of LOD (3 σ) because the intercept value acts as a constant value that does not interrupt the blank measurement and also does not affect the calculated standard deviation.

  1. The analysis of CRM’s (lines 452-459) has not been evaluated by the Student t-test to get information about the systematic error, i.e. bias, of the analytical procedure. In addition, the standard-addition method using the sample of Kiel-sea sample should be applied to get information about the recovery and thus, the effect of sample salinity could be eliminated.

Many thanks for these comments. Done. The following paragraph was addedThe paired t test was used to detect systematic error (bias) at a degree of freedom (df) of 9. No bias was observed for NO3- (t-value = - 2.46, tcritical-value (two-tails) = 3.250, p > 0.01), which is not the case for PO43- (t-value = 7.95, tcritical-value (two-tails) = 3.250, p < 0.01) and also for H4SiO4 (t-value = 6.163, tcritical-value (two-tails) = 3.250, p < 0.01), where there is a significant difference between the assigned values and the measured values. This could be due to the fact that only one CRM was tested.”

The use of Kiel Sea samples to prepare the standard solutions could not give us any advantage in correcting the salinity compared to the artificial seawater prepared in the laboratory, because the salinity variations in the Kiel Fjord are high. It was observed that the salinity fluctuated between 9 and 17 over a period of one month (May 28 – June 27).

  1. It is not clear what is the novelty of paper since the basic parameters – figures of merit (see Table 1) before optimization is missing. In addition, the LOD’s are almost the same for other FIA analyzers (see table 1).

The following paragraph was added in the conclusion to clarify the novelty of the paper as follows “The LODs for the nutrient analysis by the analyzer are indeed close to those reported in literature or for commercially available systems, but their applicability for long-term on-site monitoring of multiple nutrients in natural waters is limited by a range of drawbacks, including.

- Option to only determine a single nutrient by an analyzer.

- The use of a cadmium column for nitrate reduction, which may degrade by organic matter in the water and also regular regeneration is typically needed. Our VCl3 reduction approach therefore provides an important step forward.

- An absence of reports on long-term use or field testing in natural waters for some promising analyzers.”

  1. The careful interpretation of experimental data for phosphate (mean value 0.26 mikroM] should be taken since LOD is about 0.18 mikroM.

Many thanks for the comment. Careful interpretation of experimental data was considered in all measurements.

Reviewer 3 Report

       On-site determination of multi-nutrients are crucial for critical for studying the biogeochemistry of N, P and Si. The authors improved a commercial nutrient sensor with modified chemical methods and successfully applied it for long-term monitoring in real-world harsh environment. The field data is solid and the evaluation of the sensor is comprehensive. In all, the manuscript can be accepted after adding some important references shown below.

As one of the novelties of this work is replacement of Cd-reduction with VCl3 reduction, some important references should be cited. For example, the first flow analysis with VCl3 reduction for determining nitrate in natural water (Wang et al., Talanta, 2016, 146, 744-748), VCl3 reduction with similar flow mode of this work (Fang et al., Analytica Chimica Acta, 2019, 1076, 100-109), on-site long-term monitoring nitrate with VCl3 reduction (Nightingale et al., Environmental Science & Technology, 2019, 53, 9677-9685).

Author Response

Response to reviewers

Dear Editor,

Many thanks for handling our manuscript. We like to thank the reviewers for their inspiring words, and very much appreciate the comments by the reviewers, and have followed up their advice and amended the manuscript accordingly. Below, we present the reviewers’ comments with our replies. We hope that we have satisfied the requirements by you and the reviewers.  

Below, you will find responses to reviewers’ comments for our article entitled

Improvement of on-site sensor for simultaneous determination of phosphate, silicic acid, nitrate plus nitrite in seawater

Comments:

  1. As one of the novelties of this work is replacement of Cd-reduction with VCl3reduction, some important references should be cited. For example, the first flow analysis with VCl3reduction for determining nitrate in natural water (Wang et al., Talanta, 2016, 146, 744-748), VCl3 reduction with similar flow mode of this work (Fang et al., Analytica Chimica Acta, 2019, 1076, 100-109), on-site long-term monitoring nitrate with VCl3 reduction (Nightingale et al., Environmental Science & Technology, 2019, 53, 9677-9685).

Done. The following statement was added to the introduction “This method has been used for a decade in flow analyzers for on-site monitoring of nitrate in natural waters.” and references 36, 37 and 38 were cited.

Round 2

Reviewer 1 Report

The authors have thoroughly answered all of the remarks I raised in the first round of review and improved the manuscript according. After a very careful and exhaustive evaluation of the revised manuscript, I can finally recommend it for publication in Sensors to add knowledge to the field of nutrient detection.

The authors are also encouraged to address the following minor grammatical errors below:

Line 16, no comma after NO2-

Line 38, in the regulation of ocean productivity

Line 54, use “,” instead of “and” before expensive

Line 55, no space before yield and delete “to” after it

Line 63, [12][13] should be consistent with the rest of the text [12,13]

Line 71, space is needed after “cases” in “some cases eliminate”

Line 124, the font of “for on-site and in situ” is different than the rest of the manuscript

Line 138, “there” are instead of “they”

Line 146, remove “is reduced”

Line 268, space is needed after “blue” in “bluecolored product”

Line 298, “The solution is then injected into the detector” is a repeated sentence

Line 761, “Validations were performed”

Author Response

Dear reviewer,

Thank you very much for your corrections. Done and all minor grammatical errors have been corrected.

Reviewer 2 Report

It seems that authors have accepted the majority of my remarks. Only t-value for statistical testing should be absolute value to test the significance of parameter. I recommend this paper after minor corrections.

Author Response

Dear reviewer,

Thank you very much for your corrections. 

Done and we take the absolute value of the t-value and the sentence became as follows “No bias was observed for NO3- (t-value = 2.46, tcritical-value = 2.82, p > 0.01), which was not the case for PO43- (t-value = 7.95, tcritical-value = 2.82, p < 0.01) and H4SiO4 (t-value = 6.163, tcritical-value = 2.82, p < 0.01).”